



# Improving the stomatal resistance, photosynthesis and two big leaf algorithms for grass in the regional climate model COSMO-CLM

Evgenii Churiulin[1], Vladimir Kopeikin[2], Markus Übel[3], Jürgen Helmert[3], Jean-Marie Bettems[4], Merja Helena Tölle[1]

[1]Center for Environmental Systems Research, University of Kassel, 34117 Kassel, Germany
[2]Hydrometcenter of Russia, 123242 Moscow, Russia
[3]Deutscher Wetterdienst, 63067 Offenbach am Main, Germany
[4]Federal Office of Meteorology and Climatology, Zurich, CH-8058, Switzerland

*Correspondence to*: Evgenii Churiulin (evgenychur@uni-kassel.de)

**Abstract.** Climatic changes towards warmer temperatures require the need to improve the simplified vegetation scheme of the regional climate model COSMO-CLM, which is not capable of modelling complex processes which depend on temperature, water availability, and day length. Thus, we have implemented the physically based Ball-Berry approach coupled with photosynthesis processes based on Farquhar and Collatz models for $C_3$ and $C_4$ plants in the regional climate model COSMO-CLM (CCLM v 5.16). The implementation of the new algorithms includes the replacement of the "one-big leaf" approach by a "two-big leaf" one. We performed single column simulations with COSMO-CLM over three observational sites with $C_3$ grass plants in Germany for the period from 2010 to 2015 (Parc, Linden and Lindenberg domain). Hereby, we tested three alternative formulations of the new algorithms against a reference simulation (CCLMref) with no changes. The first formulation (CCLM3.5) adapts the algorithms for stomatal resistance from the Community Land Model (CLM v3.5), which depend on leaf photosynthesis, $CO_2$ partial and vapor pressure and maximum stomatal resistance. The second one (CCLM4.5) includes a soil water stress function as in CLM v4.5. The third one (CCLM4.5e) is similar to CCLM4.5, but with adapted equations for dry leaf calculations. The results revealed major differences in the annual cycle of stomatal resistance compared to the original algorithm (CCLMref) of the reference simulation. The largest changes in stomatal resistance are observed from October to April when stomata are closed while summer values are generally less than control values that come closer to measured values. The results indicate that changes in stomatal resistance and photosynthesis algorithms can improve the accuracy of other parameters of the COSMO-CLM model (e.g.: transpiration rate or total evapotranspiration). These results were received by comparing COSMO-CLM parameters with FLUXNET data, meteorological observations at the sites, and GLEAM and HYRAS datasets.



## 1 Introduction

The land surface processes significantly affect the conditions in the low-level atmosphere (Tölle and Churiulin 2021). The surface radiation budget and turbulent heat fluxes are controlled by near-surface atmospheric conditions. They also determine the amount of energy and water available for heat and moistening the air over land. The main parameters, which determine the interactions between the land surface and atmosphere, are the soil water content (Koster et. al., 2002) and the surface roughness (de Noblet-Ducoudre and Pitman, 2021). The impact of surface processes is evident in the low-level temperature, humidity,

the structure of the planetary boundary layer and precipitation (Arora, 2002). Tölle et al. 2014 have shown in climate simulations at convection-permitting scale that vegetation type changes can have a significant impact on extreme temperatures. Associated changes in vegetation phenology influence the energy and water cycle. Therefore, atmospheric models have to represent the land surface processes in a realistic way.

However, the evapotranspiration simulated by the multilayer land surface scheme TERRA-ML of the Consortium for Small-scale Modelling – COSMO (http://www.cosmo-model.org/, last access: 09 September 2021) was found to be systematically underestimated from April to October during the growing season. Evapotranspiration accounts for 60% (some catchments – up to 95%) of precipitated water, comprising the largest component of the terrestrial hydrological cycle (Fisher et al., 2005). Almost 80 % (Jasechko, 2013) of terrestrial evaporation accounts for transpiration maintaining a mass balance between plant

transpiration and $CO_2$ uptake. Consequently, also other components of the energy and water cycles at the surface demonstrate systematic model errors, which are manifested in the turbulent heat fluxes or the soil water content.

One of the possible reasons of the underestimation of evapotranspiration is connected with the fact that in TERRA-ML the vegetation is not sufficiently represented in the surface energy balance (Schulz et al., 2015). The plant transpiration is an

important parameter which is coupled to the carbon and water cycles and acts as principal feedback between the land surface and atmosphere (Matheny et al., 2014). However, the plant transpiration is calculated in current version of TERRA-ML with errors (Stockle, 2001), which are related to the simplified parametrization scheme for stomatal conductance ($g_{st}$) or its reciprocal – stomatal resistance ($r_s$). Stomatal conductance controls the transpiration rates and is an important variable in evaluating plant physiological response to dynamic biophysical, environmental, soil water conditions and $CO_2$ concentration

of the immediate surrounding of the leaf (Übel, 2015). The carbon uptake rates are also limited by stomatal conductance. Plants are able to balance the uptake of $CO_2$ required for photosynthesis with the need to maintain sufficient moisture levels inside the leaf (Matheny et al., 2014). Leaves close their stomata (low light level, cold temperature, high $CO_2$ volume, low leaf nitrogen, dry leaf and air) to avoid decreases in leaf water potential that could lead to desiccation or catastrophic cavitation within the xylem system (Davin et al., 2013). The open stomata (high light level, warm temperature, moderate $CO_2$ volume,

high leaf nitrogen, moist leaf and air) operate dynamically and regulate water loss and C uptake (Ball, 1988). The increase in global and regional temperatures and potential rise of the variation in regional precipitation creates the necessity for more accurate prediction of stomatal conductance (resistance) depending on heat, water and carbon exchange (Berry et al., 2010;





Wu et al., 2012; Jasechko et., 2013). Nevertheless, the current parametrization scheme of TERRA-ML for stomatal conductance does not take into account the stomatal regulation and vegetation growth interacting with atmospheric $CO_2$.

Moreover, the LAI does not respond to water stress, and depends on vegetation parameters. Also, the COSMO-CLM applies the "one-big-leaf" approach for radiation fluxes, although this approach has disadvantages that are related to the impossibility of accounting for the difference of the physiological properties between sunlit and shaded leaves (Dai et al., 2004), and overestimation of the reduction of photosynthesis ($A$) when clouds attenuate solar radiation (Übel, 2015).

Most dynamic vegetation models use plant functional types (PFT) as a source of vegetational parameters and a tile approach. The complex phenology and photosynthesis schemes exist in dynamic vegetation models (CARAIB (Dury et al., 2011), JSBACH (Reick et al., 2021), the Community Land Model (CLM) (Oleson et al., 2010 and 2013), the Lund-Potsdam-Jena-GUESS (LPJ-GUESS) model (Smith et al., 2014), the ORCHIDEE (Krinner et al., 2005)). However, these schemes have not been implemented into production (exploitation) at convection-permitting scale (Prein et al. 2015), either in COSMO-CLM or

in ICON-CLM (Giorgetta et al., 2018) yet. There are several successful examples of the CLM implementation into different regional climate models, for example in WRF model (Van Den Broeke et al., 2017) or in COSMO-CLM (Davin et al., 2011; Davin and Seneviratne, 2012). The last version is called COSMO-CLM[2] and the main idea of this version is coupling to different models: COSMO-CLM (v4.8) and CLM3.5. Davin et al. (2011) have coupled COSMO-CLM with CLM and found improvements with respect to land surface fluxes, including an improved magnitude of radiation fluxes and a better partitioning

of turbulent fluxes. It should be noted that the soil model (TERRA-ML) used in the COSMO-CLM was fully replaced in COSMO-CLM[2] with the CLM3.5 parametrization scheme. The COSMO-CLM[2] was created and tested, but Davin et al. (2011) did not perform the convection-permitting scale simulations (Prein et al. 2015), because the computational costs were high (Stökli et al., 2008 and 2011).

In order to overcome these limitations, we decided to substitute the empirical Jarvis approach with the physically based Ball-Berry (Ball, 1988; Ball and Berry, 1991) approach coupling with photosynthesis (Farquhar et al. 1980 and Collatz et al., 1991 models for $C_3$ and $C_4$ plants) and introduced a "two-big leaf" canopy (Thornton and Zimmermann, 2007). All our improvements were directly implemented in TERRA-ML of COSMO-CLM that allowed us to improve TERRA-ML and save all dignity of COSMO-CLM (for example: convection-permitting scale). These changes distinguish our research from the research of (Davin

et al., 2011; Davin and Seneviratne, 2012) for coupling COSMO-CLM and CLM models.

The implementation of the new algorithms to COSMO-CLM required a lot of work, which was related to technical aspects (e.g.: COSMO-CLM model uses land use classes (LUC) as a source of vegetation parameters, while Ball-Berry approach for calculating stomatal resistance requires vegetation parameters from PFTs). For the first step of our research, we decided to

simplify this technical part of the study and work with only one plant type – grass. This assumption allowed us to implement and verify the new algorithms for stomatal resistance, leaf photosynthesis, and "two-big leaf" and prepare the basis for the implementation of the new PFT which do not coincide with COSMO-CLM land use classes.



In section 2, the modelling system COSMO-CLM is described as well as the phenology algorithms of the current version of TERRA-ML, the substantial improvements of TERRA-ML and COSMO-CLM. The model experiments, selected domains and
observational data used for verification are presented in Section 3. In section 4, the control, experimental and statistical results are presented and discussed. The results are summarized in Section 5.

## 2 Methods

### 2.1 Model description

COSMO-CLM is a climatic version (Rockel et. al., 2008) of a nonhydrostatic limited area atmospheric prediction model of
the Consortium for Small-Scale Modelling (COSMO). COSMO is applied for both meso-$\beta$ and meso-$\gamma$ scales. The model has a system of horizontal (rotated) and vertical (terrain-following height) coordinates (Doms et al., 2018). There is a scheme of a moist atmosphere (fully compressible) in the model based on non-hydrostatic thermo-dynamical equations, which are solved numerically on an Arakawa-C staggered grid (Arakawa and Lamb 1977) with a Runge-Kutta time-stepping scheme (Wicker and Skamarock 2002). The precipitation parameterization scheme uses a one-moment microphysics scheme for five categories
of hydrometeors (cloud, rain, snow, ice and graupel). In COSMO, we work with different kinds of parametrizations of turbulent kinetic energy-based surface transfer and planetary boundary layer (Raschendorfer 2001). The radiative transfer scheme (Ritter and Geleyn, 1992) is also applied. The surface and soil processes are calculated in the multi-layer soil model TERRA-ML (Schrodin and Heise 2002).

The soil model consists of two parts: the first part is related to computation of atmospheric parameters under the soil and the second one considers hydrological processes including snow melting and freezing. In our version of the soil model, we are using ten (up to 15.34 m depth) and eight (up to 3.82 m depths) active layers for energy and water transport calculations. The TERRA-ML parameterizes all surface fluxes at a grid element and sums them up into a total moisture flux – evapotranspiration. The transpiration from vegetation is calculated for vegetated areas. The bare soil evaporation is computed for nonvegetated
areas. Plant transpiration and bare soil evaporation are not considered for ice and rock soil types. For other soil types, the calculations are based on the Biosphere-Atmosphere Transfer Scheme – BATS (Dickinson et al., 1993). The BATS-based formulation of the plant transpiration is presented in Eq. (1):

$$T_r = f_{plnt} \left(1 - f_i\right) \left(1 - f_{snow}\right) E_{pot}\left(T_{sfc}\right) r_a \left(r_a + r_f\right)^{-1} \tag{1}$$

where $Tr_k$ is plant transpiration, $r_a$ and $r_f$ are atmospheric and foliage resistance, $f_{plnt}, f_i, f_{snow}$ are fractional area covered
by plants, interception water and snow, $E_{pot}\left(T_{sfc}\right)$ is potential evapotranspiration. As it was previously mentioned, the foliage resistance is related to leaf area index and describes the reduction of transpiration by stomatal resistance ($r_s$). The current formulation of stomatal resistance is presented in next Section 2.2.



## 2.2 Stomatal resistance

### 2.2.1. Current formulation

In the current model version of TERRA-ML, the stomatal resistance computations are based on the multiplicative and simple resistance Jarvis-Stewart approach (Jarvis, 1976; Stewart, 1988) with the BATS model parameterization (Dickinson et al., 1993) – Eq. (2).

$$r_s^{-1} = r_{\max}^{-1} + (r_{\min}^{-1} - r_{\max}^{-1})[F_{rad}\,F_{tem}\,F_{wat}\,F_{hum}], \tag{2}$$

where $r_{min}$ is minimum stomatal resistance from external dataset, which varies depending on the land cover type, $r_{max}$ is
maximum stomatal resistance equal to 4000 s/m. The functions $F_{rad}$, $F_{tem}$, $F_{hum}$, $F_{wat}$ are stress environmental functions which are related to photosynthetic active radiation, ambient temperature, ambient specific humidity, and soil water content. The limitation functions take values in the range from zero (unfavourable conditions) to one (optimum conditions). As the stomatal resistance is linked to near-surface temperature and soil water content, it also depends on applied physiographic data in the model (Smiatek et al., 2016).

The Jarvis-Stewart is phenomenological approach, which is based on empirical dependencies between canopy resistance ($r_{can}$) and environmental variables. This approach has disadvantages. It is an empirical approach with statistical dependencies used for determining the model parameters for different plant types. The functions which are applied in Jarvis approach are independent of each other (Collatz et al., 1991). The method does not take into account the influence of atmospheric $CO_2$
concentration. The prognostic environmental parameters are functions of stomatal resistance as they are (Ronda et al., 2001). Moreover, COSMO-CLM applies highly simplified dependencies, for which the leaf photosynthesis and $CO_2$ uptake cannot be calculated, and the canopy layer is represented as a "big-leaf". The detailed description of COSMO-CLM and TERRA-ML can be found in (Doms and Baldauf, 2018; Doms et al., 2018; Schraff and Hess 2018; Schättler et al., 2018) and the official webpage of the COSMO consortium (http://www.cosmo-model.org/, last access: 09 September 2021).

### 150 2.2.2. New formulation

The new description of the stomatal resistance in TERRA-ML is calculated on the basis of the plant physiological approach (Ball et al, 1987; Ball, 1988) with algorithms for canopy fluxes based on Collatz model (Collatz et al., 1991) and improved by (Thornton and Zimmermann, 2007) through the implementation of a new parametrization scheme for the maximum rate of carboxylation ($V_{c,max}$) which was the most critical problem of Collatz model. The main principle of Collatz model is that
stomatal resistance ($r_s$) depends on the environmental conditions and allows to relate stomatal conductance (the inverse of resistance) to net leaf photosynthesis, scaled by relative humidity and the $CO_2$ concentration. In our research, we applied two different algorithms for calculations of stomatal conductance depending on the two different CLM versions 3.5 (Oleson et al., 2010) and 4.5 (Oleson et al., 2013) which are presented in Eq. (3) and Eq. (4):



$$g_{st} = \frac{1}{r_s} = m \frac{A_n}{c_s} \frac{e_s}{e_i^*} P_{atm} + b, \qquad (3)$$

$$g_{st} = \frac{1}{r_s} = m \frac{A_n}{c_s} \frac{e_s}{e_i^*} P_{atm} + b\beta_t, \qquad (4)$$

where $r_s$ is leaf stomatal resistance [s m$^2$ μmol$^{-1}$], $A_n$ is leaf photosynthesis [μmol $CO_2$ m$^{-2}$ s$^{-1}$], $e_s$ is vapor pressure at the leaf surface [Pa], $c_s$ is $CO_2$ partial pressure at the leaf surface [Pa], $e_i^*$ is saturation vapor pressure [Pa] inside the leaf at the vegetation temperature $T_v$ [K], $P_{atm}$ is atmospheric pressure [Pa], $m$ is plant functional type dependent parameter [-], $\beta_t$ is soil water stress function (which ranges from 0 to 1), $b$ is minimum stomatal conductance [μmol m$^{-2}$ s$^{-1}$]: in case of CLM 3.5  $b =$

2000;  for CLM 4.5 $b = 10000$ and $40000$ for C$_3$ and C$_4$ plants. Applying these values of minimum stomatal conductance is necessary for obtaining the maximum stomatal resistance values when $A = 0$. The implementation of the physical Ball-Berry approach required the implementation of the new algorithms for leaf photosynthesis (Section 2.3) and radiation (Section 2.4).

### 2.3 Leaf photosynthesis

### 2.3.1. Current formulation

In the current model version of TERRA-ML, there are no algorithms for the computation of leaf photosynthesis.

### 2.3.2. New formulation

The new description of the leaf photosynthesis Eq. (5) of TERRA-ML is based on the Farquhar and Collatz (Farquhar et al. 1980 and Collatz et al., 1991) models for C$_3$ and C$_4$ plants and uses sunlit and shaded leaves parameters for calculations:

$$A_n = A^{sun} L^{sun} + A^{sha} L^{sha}, \qquad (5)$$

where $A^{sun}$ and $A^{sha}$ are sunlit and shaded values of leaf photosynthesis, $L^{sun}$ and $L^{sha}$ are LAI for sunlit and shaded leaves. According to the CLM (Oleson et al., 2010 and 2013) strategy, the minimum rate set by one of the limitation relations controls $CO_2$ assimilations at the leaf level Eq. (6).

$$A^{sun,sha} = \min (w_c, w_j, w_e), \qquad (6)$$

where $w_c, w_j, w_e$ are limitation factors, which are related to the rate of $CO_2$ fixation in the carboxylation of RuBP in the Calvin

cycle [μmol $CO_2$ m$^{-2}$ s$^{-1}$], the maximum rate of carboxylation allowed by the capacity to regenerate RuBP [μmol $CO_2$ m$^{-2}$ s$^{-1}$], and the capacity for the export or utilization of the carbohydrates [μmol $CO_2$ m$^{-2}$ s$^{-1}$]. The algorithm for calculation of the limitation factors (according to CLM model) is presented in (Appendix A).





### 2.4 Radiation fluxes

#### 2.4.1. Current formulation

In the current model version of COSMO-CLM the canopy layer is presented as a "one-big leaf" having the same plant physiological properties and environmental controls as all leaves of the canopy (Doms et al., 2018).

#### 2.4.2. New formulation

The new description of the canopy layer of COSMO-CLM is presented as a "two-big leaf". Due to the implementation of photosynthesis and stomatal resistance algorithms we had to substitute "one-big leaf" approach with a more modern "two-big

leaf" one. The main difference in methods is that the "two-big leaf" approach contains sunlit and shaded leaves and allows to calculate stomatal resistance and photosynthesis rate more accurately, considering different leaves properties. Sunlit leaves receive (absorb) beam direct and diffuse solar radiation. Shaded leaves receive (absorb) only scattered diffuse solar radiation. Applying the "two-big leaf" approach allows to calculate the LAI separately for sunlit Eq. (7) and shaded Eq. (8) leaves, which is necessary for leaf photosynthesis and stomatal resistance calculations.

$$L^{sun} = f_{sun}L, \tag{7}$$

$$L^{sun} = f_{sun}L, \tag{8}$$

where $L$ is leaf area index, $f_{sun}$ and $f_{sha}$ are sunlit and shaded leaves fractions which vary over the course of a day and through the year. Furthermore, the implementation of the "two-big leaf" approach demanded creation of the new algorithms for calculating the specific leaf area (SLA) indexes for sunlit Eq. (9) and shaded Eq. (10) leaves, which can be used to estimate

the reproductive strategy of a particular plant.

$$SLA^{sun} = \frac{-(cSLA_mKL + cSLA_m + cSLA_0 - SLA_m - SLA_0K)}{L^{sun}}, \tag{9}$$

$$SLA^{sha} = \frac{L\left(SLA_0 + \frac{SLA_mL}{2}\right) - SLA^{sun}L^{sun}}{L^{sha}}, \tag{10}$$

where $SLA_0$ is value for SLA at the top of the canopy, $SLA_m$ is linear slope coefficient, $c$ is recalculation coefficient. Applying the "two-big leaf" approach improve the accuracy of photosynthesis and transpiration calculations and connect them with

structural ($SLA_m, L^{sun/sha}$) and functional ($SLA^{sun/sha}$) characteristics of the canopy. For our research we applied the free-available the CLM3.5 source code (https://www.earthsystemgrid.org/dataset/ucar.cgd.clm.src.3.5.src.html, last access: 09 September 2021) and the official model documentations for versions CLM 3.5 and 4.5 (Oleson et al., 2010 and 2013).

### 2.5 Statistical analysis

In our study, we analysed the data in two different ways: the first one is the analysis of the data near the meteorological stations.

The model results and the data from the HYRAS and GLEAM datasets were averaged to one point (station) based on the four



closest to the station model grid points. In that case, the hourly and daily values are evaluated by the standard deviation (STD), the mean absolute error (MAE), the root mean square error (RMSE) and the Pearson correlation coefficient (PCC).

The second method was analysing the experiments, which are presented on the COSMO-CLM model grid. Hereby, the air
temperature variables (two meter – $T_{2m}$, surface – $T_S$, maximum – $T_{max}$ and minimum – $T_{min}$) of the simulations are compared to the HYRAS observational dataset. The other two variables of COSMO-CLM (the total evapotranspiration – ZVERBO and the amount of water evaporation – AEVAP) are compared with the GLEAM datasets. The HYRAS and GLEAM parameters were tentatively interpolated on COSMO-CLM domain grids. For this analysis, we calculated: 1 – the PCC, which reflects the quality and the spatial consistency of the simulations and observations. 2 – the distribution added value index – DAV (Eq.
(11)) to estimate the Perkins skill scores (S) between reference (subscript – ref), experimental (subscript – exp), and observational (subscript – obs) data; 3 – the Kling-Gupta Efficiency index – KGE (Eq. (12)) to demonstrate the model (subscript – m) effectiveness with respect to the observational time-series and 4 - the root-mean-square deviation (RMSD).

$$DAV = \frac{\sum_1^n \min(Z_{exp}, Z_{obs}) - \sum_1^n \min(Z_{ref}, Z_{obs})}{\sum_1^n \min(Z_{ctr}, Z_{obs})},$$
(11)

where $Z$ is frequency of experimental, referential, and observational values in a specific bin. The values of DAV > 0 show that
there is a benefit in using the alternative experiment version compared to the reference with respect to the observations, DAV ≤ 0 indicates that there is either no gain or we have a loss in performance for the alternative version (Raffa et al., 2021).

$$KGE = 1 - \sqrt{(\rho - 1)^2 + \left(\frac{\sigma_m}{\sigma_{obs}}\right)^2 + \left(\frac{\mu_m}{\mu_{obs}} - 1\right)^2},$$
(12)

where $\sigma$ is standard deviation, $\mu$ is mean value, $\rho$ is the Pearson correlation coefficient. The values of KGE < -0.41 demonstrate that there is a lack of precision in relation to the mean of the control (observational) data. KGE = 1 indicates that
there is a perfect matching between experimental and control data (Tölle and Churiulin, 2021).

## 3 Experiments and observational data

### 3.1 Model experiments

In carrying out our research, we have implemented the new algorithms in COSMO-CLM v5.16. The first presented version (CCLMv3.5) is based on CLM v3.5 algorithm for stomatal resistance, which depends on leaf photosynthesis, $CO_2$ partial and
vapor pressure and maximum stomatal resistance. The second one (CCLMv4.5) is based on the phenology algorithms of CLM v4.5 including the soil water stress function. The third one is similar to the experiment (CCLM v4.5), but with adapted equations for dry leaf calculations (CCLMv4.5e). The algorithms were tested only for $C_3$ grass. The main reason of it is related to the adaptation of plant functional types (PFT) to COSMO-CLM land use classes (LUC). In the case of $C_3$ grass these approaches are similar. In COSMO-CLM, we have implemented two additional modules with the new algorithms and added





the new variables (for example: stomatal resistance of sunlit – RSTOM$_{sun}$ and shaded – RSTOM$_{sha}$ leaves, leaf photosynthesis – PSN). Moreover, the radiation module of COSMO-CLM was changed. The new algorithm for "two-big leaf" approach was added to the radiation part of COSMO-CLM. The implementation of the "two-big leaf" approach demanded introducing the new parameters in the radiation part of COSMO-CLM model for extracting direct component ($\phi_{dir}^{\mu}$ [W m$^{-2}$]), diffuse downward component ($\phi_{dif}^{\mu}$ [W m$^{-2}$]), and diffuse upward component ($\phi_{dif}$ [W m$^{-2}$]) of photosynthetic active radiation at the

ground for sunlit ($\phi^{sun}$ –Eq. (13)) and shaded ($\phi^{sha}$ – Eq. (14)) leaves.

$$\phi^{sun} = \frac{\left(\phi_{dir}^{\mu} + \phi_{dif}^{\mu}f_{sun} + \phi_{dif}f_{sun}\right)\left(\frac{L}{L+S}\right)}{L^{sun}} \; for \; L^{sun} > 0, \tag{13}$$

$$\phi^{sha} = \frac{\left(\phi_{dif}^{\mu}f_{sha} + \phi_{dif}f_{sha}\right)\left(\frac{L}{L+S}\right)}{L^{sha}} \; for \; L^{sha} > 0, \tag{14}$$

The results of the experiments made on the basis of the one-dimensional version of the regional climate model COSMO-CLM are combined and presented in the vertical soil-vegetation-atmosphere column. This set-up allows to study surface and

vegetation exchange processes and avoid the large-scale atmospheric effects. The one-dimensional version also allows for easy interpretation of the results on vegetation-atmosphere interactions because the horizontal advection can be ignored.

The COSMO-CLM experiments (CCLMv3.5, CCLMv4.5, CCLMv4.5e) in a one-column mode have been analysed and compared: 1 – to the control simulation (CCLMref) with the original algorithm, 2 – with observational sites (in the cases when

the observational data was available) and values for stomatal resistance out of the literature (only for stomatal resistance), 3 – available datasets (EURONET, HYRAS and GLEAM). In our research we analysed the climatological annual and daily cycles of the model variables from 2010 to 2015 for three small study domains (Fig. 1a) with mixed grass biome types (the "Parc" with the coordinates 50.8N – 50.9N × 6.38E – 6.60E; the "Linden" 50.2N – 50.8N × 8.4E – 8.8E and the "Lindenberg" 52.2N – 52.4N × 14.0E – 14.4E). Mixed means that grass was combined with or surrounded by crops.





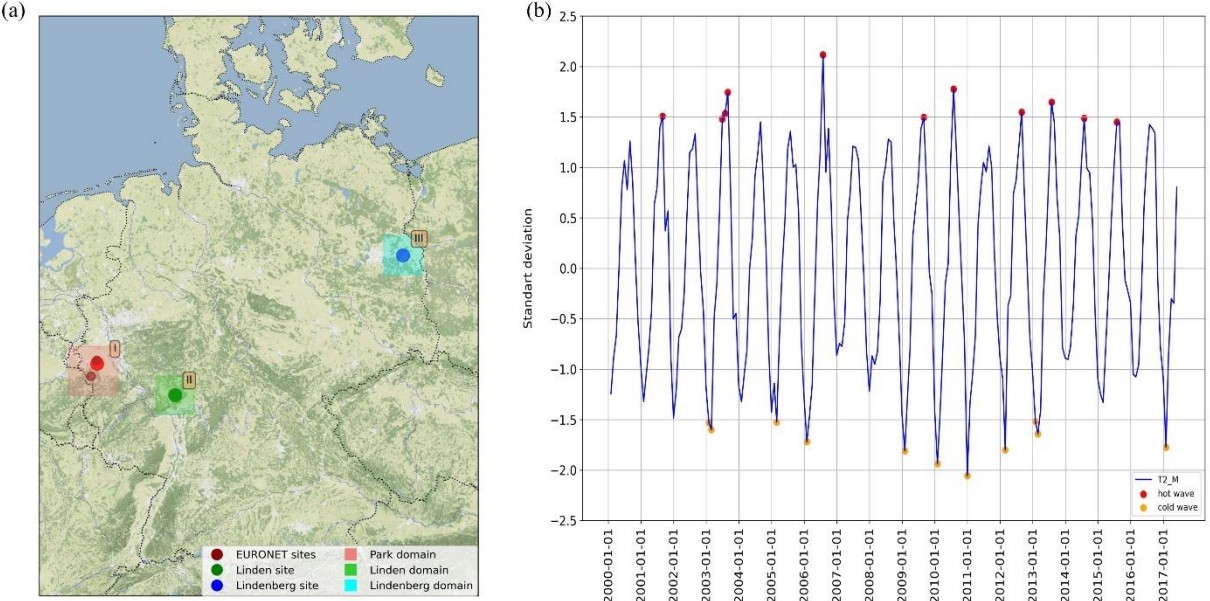

**Figure 1: The sites and research COSMO-CLM domains over Germany (a); the standardized anomalies from 2000 to 2017 on the research domains (b). An initial layer with information about altitudes for the map (a) was obtained from the official webpage of NOAA (https://www.ngdc.noaa.gov/mgg/global/ last access: 30 September 2021)**

The focus of the research was on statistical analysis of the summer months when the vegetation is in active phase. The greatest

interest for comparison presents parameters, which are related to the energy balance components, or the parameters that influence them, for example: surface temperature, the latent and sensible heat fluxes, the fluxes of water evaporation and transpiration, stomatal resistance. The time-period for COSMO-CLM simulation is from 1999 to 2017 and from 2010 to 2015 for our experiments. The model time-step between calculations is 15 seconds and the simulation output of the most parameters is equal to one hour. The experiment period from 2010 to 2015 was defined because of several reasons: 1 – the in-situ data for

most of EURONET data in our research domains are available from 2011 to 2020; 2 – the period includes the highest number of standardized anomalies for each 5 years Eq. (15) for all research domains from 1999 to 2017 (Fig. 1b).

$$T_{2m, \ stand} = \frac{T_{2m} - T_{2m, \ mean}}{T_{2m, \ std}} \ , \tag{15}$$

where $T_{2m}$ is air temperature at 2m, $T_{2m, \ mean}$ is mean air temperature at 2m, $T_{2m, \ std}$ is a standard deviation of $T_{2m}$.

### 3.2 In-situ data

For this study, three EURONET sites were selected; two cropland sites (Selhausen Juelich and Selhausen) and one grassland site (Rollesbroich). Although, all sites are surrounded either by crops or grass. All sites are located in the western part of Germany near the Belgian border. We also used grassland data from "Environmental Monitoring and Climate Impact Research Station Linden" which is located near Giessen and data from "Meteorologisches Observatorium Lindenberg" near Berlin. The



sites were selected in such a way that they cover similar types of plants (C$_3$ grass) in different parts of Germany. In addition
to this, the data have been available since 2007 and, by preference for example, include 2015, during which a heat and drought
wave was observed in Germany. Data availability for each site is provided in Table 1. Sites and corresponding COSMO-CLM
domains are shown in (Fig. 1a). Forest sites were not taken into account at this stage of the research, as algorithms for C$_3$ grass
only have been implemented in COSMO-CLM model calculations.

**Table 1: Metainformation on the sites and flux towers used to verify COSMO-CLM experiments**

| Site | Location | | Elev. | Climate | | Biome type | Webpage | Available data |
|---|---|---|---|---|---|---|---|---|
| | Lat($^o$) | Lon($^o$) | (m) | T (°C) | P (mm) | | | |
| Selhausen Juelich | 50.86 | 6.45 | 101 | 10.0 | 700 | Croplands | http://www.europe-fluxdata.eu/ last access: 03 June 2021 | 2011 – 2020 |
| Selhausen | 50.87 | 6.44 | 103 | 9.9 | 693 | Croplands | | 2007 – 2010 |
| Rollesbroich | 50.62 | 6.30 | 511 | 7.7 | 1033 | Grass | | 2011 –2020 |
| Linden | 50.32 | 8.41 | 172 | 10.3 | 819 | Grass | https://www.uni-giessen.de last access: 06 June 2021 | 2004 – 2011 |
| Lindenberg | 52.10 | 14.07 | 73 | 10.4 | 790 | Grass | https://www.dwd.de/ last access: 06 June 2021 | 2007 – 2020 |

**3.3 Datasets (E-OBS, HYRAS, GLEAM)**

As additional resources of data for validating COSMO-CLM, we applied gridded datasets with information about precipitation,
temperature and evaporation. The datasets have parameters of grids, which are different from the COSMO-CLM grid. The
datasets which are presented on their grids were interpolated on COSMO-CLM grid that allowed us to get more data that are
reliable for comparison. The three sets of COSMO-CLM grid parameters were prepared for the interpolation of datasets on
COSMO-CLM grids (one set for one COSMO-CLM domain). Comparison of the COSMO-CLM data with datasets on the
interpolated grids is an important instrument for validating the model data, as COSMO-CLM demonstrates similar spatial scale
results of processes (model data represent average values) rather than processes in specific points (Osborn and Hulme, 1998).

The E-OBS data set was developed as part of the European Union Framework 6 ENSEMBLES project (Cornes et al., 2018)
for validating Regional Climate Models (RSM) and for climatic change studies (Haylock et. Al., 2008). This data set contains
daily values of temperature (mean, minimum and maximum) and precipitation. The spatial resolution of the gridded data is
available on a 0.1-degree regular grid. The gridded data set was formed from the interpolation of station-derived meteorological
measurements. The standard deviation for temperature values is 3.2 °C and 1.53 mm for precipitation (Cornes et al., 2018).

The HYRAS data is a high-resolution gridded data set (5 km × 5 km) with various hydrometeorological parameters for
Germany and the bordering river catchment. The data set was developed by DWD and is updated once every five years (Frick
et.al., 2014). We applied HYRAS data from 1999 to 2016 with daily values of temperature (mean, minimum and maximum)





and relative humidity. The annual mean absolute error for temperatures is: $T_{mean} = 0.68$ °C, $T_{min} = 0.76$ °C, $T_{max} = 0.70$ °C and 4.45% for precipitation (Razafimaharo et al., 2020).


The GLEAM data is a set of algorithms dedicated to the estimation of terrestrial evaporation and root-zone soil moisture from satellite data. The GLEAM project includes two different datasets. The first one is the GLEAM v3.5a – based on global satellite and reanalysis data (ERA5). The second one is the GLEAM v3.5b – based on global satellite data (Martens et al., 2017). Both data sets contain different daily parameters of surface evaporation including transpiration, bare soil evaporation,

interception loss and sublimation (Miralles et al., 2011). The GLEAM data were validated against 91 eddy-covariance towers and 2325 soil moisture sensors across a broad range of ecosystems. More information about E-OBS, HYRAS and GLEAM datasets is available in Table 2.

**Table 2: Metainformation on the datasets used to verify COSMO-CLM experiments**

| Dataset | Coverage area | Period | Webpage | Reference | Parameters |
|---|---|---|---|---|---|
| E-OBS | 25N – 71.5N × 25W – 45E | 1950 - 2015 | https://www.ecad.eu/download/ ensembles/download.php, last access: 03 September 2021 | Cornes et al., 2018 | Mean, maximum and minimum air temperatures ($T_{mean}$, $T_{max}$, $T_{min}$) |
| HYRAS | Germany | 1950 - 2015 | https://www.dwd.de/DE/ leistungen/hyras/hyras.html, last access: 03 September 2021 | Razafimaharo et al., 2020 | Mean, maximum and minimum air temperatures ($T_{mean}$, $T_{max}$, $T_{min}$) and total precipitation ($TOT_{PREC}$) |
| GLEAM 3.5a | Global | 1980 - 2020 | https://www.gleam.eu/, last access: 03 September 2021 | Miralles et al., 2011 | Evaporation (actual – E, potential – $E_p$, soil – $E_b$, open-water – $E_w$), transpiration – $E_t$, interception loss – $E_i$, snow sublimation – $E_s$, evaporative stress – S, soil moisture (root-zone – $Sm_{root}$, surface – $Sm_{surf}$) |
| GLEAM 3.5b | | 2003 - 2020 | | | |

## 4 Results and discussion

### 4.1 Stomatal resistance

The experiment results of stomatal resistance for parc domain are shown in (Fig. 2) as climatological annual cycles (Fig. 2a – daily mean values, Fig. 2b – day values, Fig. 2c – night values) averaged over 2010 to 2015 years. The results demonstrate that the main differences between the control simulation (CCLMref) and the experiment simulations (CCLMv3.5, CCLMv4.5 and CCLMv4.5e) are observed from September (October) to March (April), when stomata are closed or there are no leaves.

Nevertheless, the changes in stomatal resistance in this period do not have a considerable influence on other COSMO-CLM variables (surface temperature, fluxes of latent and sensible heats). The main reason for this is vegetation being in inactive period.





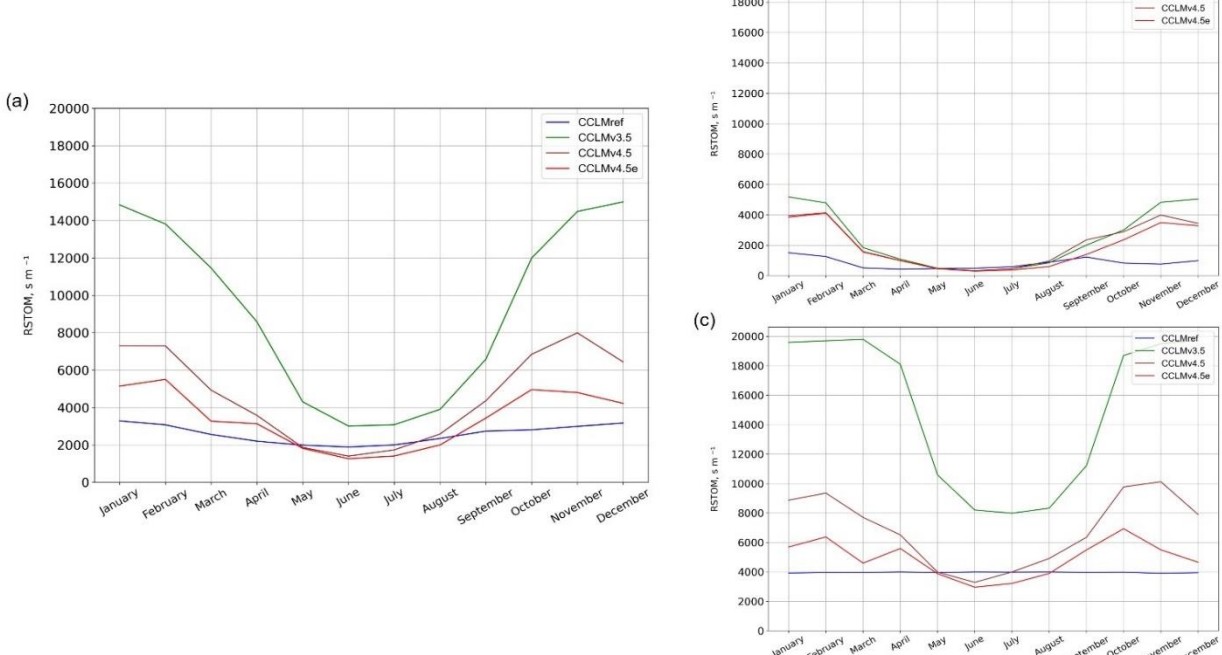

**Figure 2: Mean seasonal cycle of stomatal resistance for the parc domain based on: (a) – monthly values, (b) – values at 13:00, (c) – values at 01:00. Climatological means for CCLMref (blue line), CCLMv3.5 (green line), CCLMv4.5 (brown line) and CCLMv4.5e (red line) are calculated for the period 2010–2015**

Values of stomatal resistance vary according to changes in the environmental conditions, $CO_2$ concentration and soil water availability. The analysis of annual daily mean stomatal resistance data (Fig 2a) demonstrates that the reference experiment – CCLMref demonstrates the values which generally change from 2000 s/m (summer months) to 4000 s/m (winter months). The CCLMref values of stomatal resistance which are equal to 4000 s/m are the upper limits values for the current algorithm of TERRA-ML. According to the CCLMref all night values of stomatal resistance (Fig. 2c) are equal to 4000 s/m, regardless of season or environmental conditions. Further experiments are based on the algorithms which include the role of leaf photosynthesis, $CO_2$ concentration, and environmental parameters (temperature, humidity, soil water, active radiation) which are related to each other. The experiment – CCLMv3.5 shows the biggest fluctuations of stomatal resistance from 15000 s/m (winter) to 2500 s/m (summer). The data of the CCLMv3.5 differ significantly from the other experiments. The main reason for these big values is related to the upper limit of stomatal resistance which is set to 20000 s/m for this experiment (Oleson et al., 2010). The CCLMv3.5 reaches the maximum values daily at night when the values of leaf photosynthesis are equal to zero ($A_{night} = 0$), which is consistent with Ball's theory (Ball, 1988). The upper limit the CCLMv3.5 reaches during winter months and changes during the period with vegetation being in active. This fact is most brightly displayed in (Fig. 2c) where the night values of stomatal resistance are presented. Nevertheless, we assume that the stomatal resistance values of this experiment are overstated, especially in wintertime. The experiment CCLM_v4.5 has additional updates (including: the soil water limitation function and different upper limits for $C_3 = 10000$ s/m and $C_4 = 40000$ s/m grass). Experiment CCLM_v4.5e has the values





of stomatal resistance which are the closest to the CCLMref. CCLM_v4.5e includes the changes in stomatal resistance influenced by changes in environmental conditions, changes in diurnal cycle (similar to CCLMv4.5). In addition, algorithms

for dry and wet leaves have been implemented in COSMO-CLM from CLM 4.5. The values of stomatal resistance at 13:00 for all experiments look similar, however the experiments have lower values from May to October and bigger values from October to April that fits in well with changes in seasons, environmental conditions, values of leaf photosynthesis, and $CO_2$ concentration. Nevertheless, the annual cycle of stomatal resistance does not allow to compare the experiment values with the in-situ data.


Measuring stomatal resistance (conductance) is a resource-intensive task, especially for its continuous quantification over time, because it is a highly intermittent phenomenon, extremely localized on the leaf level, and varies with leaf positioning on a plant and from leaf to leaf and from plant to plant. There are no long in-situ time series or datasets with the daily information about stomatal resistance. We analysed different resources and found the measured in-situ stomatal resistance data, which were

published in the literature (Alfieri et al., 2008; Irmak and Mutiibwa, 2009). We compared (Fig. 3) the daily model stomatal resistance values (at 13:00 p.m. CET) with the data from the articles. It should be noted that the in-situ stomatal resistance data (only $C_3$ grass) were measured for the research domain which is located in the North America region (at 13.00 p.m. PT). We assume that our experimental stomatal resistance data for $C_3$ grass can be compared with these in-situ data.

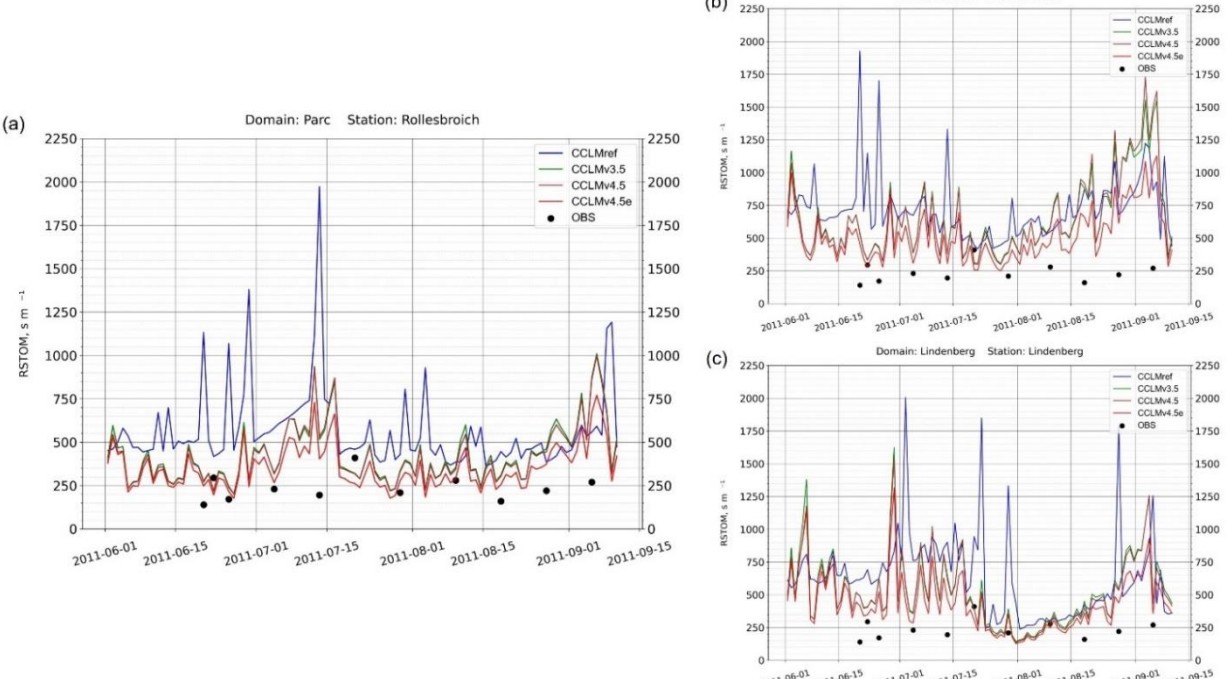

**Figure 3: Mean daily values of stomatal resistance over (a) Parc, (b) Linden and (c) Lindenberg research domains from 01.06.2011 to 15.09.2011 at 13:00 based on COSMO-CLM experiments (CCLMref – blue line, CCLMv3.5 – green line, CCLMv4.5 – brown line, CCLMv4.5e – red line). The in-situ measurements of stomatal resistance for $C_3$ grass are based on the literature review (black dots).**



The results demonstrate that COSMO-CLM values of stomatal resistance at 13:00 p.m. CET for three domains Germany are
similar to the North American stomatal resistance data (for $C_3$ grass). The CCLMref demonstrates the biggest values of
stomatal resistance at 13:00 p.m. The one possible reason for these results of CCLMref can be related to the lower limit of
stomatal resistance which depends on the plant types. The calculation results of CCLMref cannot be fewer than this lower
limit (for $C_3$ grass equal to 150 s/m). The experimental versions do not have this lower limit that allows them to be more
accurate, as confirmed with the in-situ data (Alfieri et al., 2008; Irmak and Mutiibwa, 2009). The stomatal resistance values of
the experiments have better correlation with the published data than control experiment CCLMref. The average statistical
analysis of the data from 20.06.2011 to 05.09.2011 is presented in Table 3.

**Table 3: The comparison of the results of stomatal resistance experimental data with in-situ measurements**

| Date | COSMO-CLM experiments | | | | OBS | Date | COSMO-CLM experiments | | | | OBS |
|---|---|---|---|---|---|---|---|---|---|---|---|
| | CTR | v3.5 | Date | v4.5e | | | CTR | v3.5 | Date | v4.5e | |
| 20/06/2011 | 1134 | 276 | 276 | 249 | 140 | 18/08/2011 | 446 | 310 | 310 | 255 | 160 |
| 22/06/2011 | 418 | 220 | 220 | 195 | 295 | 27/08/2011 | 386 | 462 | 462 | 373 | 221 |
| 25/06/2011 | 1069 | 246 | 246 | 217 | 172 | 05/09/2011 | 558 | 871 | 871 | 670 | 270 |
| 04/07/2011 | 582 | 317 | 317 | 267 | 230 | *mean* | 713 | 384 | 384 | 314 | 235 |
| 13/07/2011 | 1973 | 516 | 516 | 403 | 196 | *std* | 495 | 183 | 183 | 133 | 76.5 |
| 20/07/2011 | 459 | 321 | 321 | 262 | 410 | *mae* | 477 | 179 | 179 | 124 | |
| 29/07/2011 | 430 | 337 | 337 | 278 | 210 | *rmse* | 696 | 235 | 235 | 161 | |
| 09/08/2011 | 386 | 353 | 353 | 289 | 280 | *pcc* | -0.426 | 0.103 | 0.103 | 0.08 | |

## 4.2 Evapotranspiration and evaporation

According to (Davin and Seneviratne, 2012) there is a tight coupling between photosynthesis and transpiration. It's a fact that
transpiration is the main contributor to land evapotranspiration (Matheny et al., 2014). Stomatal resistance is also expected to
affect water fluxes (Übel, 2015). We analysed the data from COSMO-CLM which are related to the total evapotranspiration
(model variable – ZVERBO) and the amount of water evaporation (model variable – AEVAP) to examine the sensitivity of
these parameters to stomatal resistance changes. Additionally, we applied the GLEAM datasets (v3.5a and v3.5b) to compare
experimental results with in-situ measurements. The experimental results over three study regions for ZVERBO and AEVAP
parameters are shown in (Fig. 4) as climatological annual cycles averaged over 2010 to 2015 years. The modelling results of
these two parameters have fewer changes than in stomatal resistance calculations between experiments. Nevertheless, there
are enough of them to understand how the changes in stomatal resistance algorithms are reflected in water fluxes. The
differences between experiments from October to April are insignificant, confirming the assumption that the changes in
stomatal resistance in this period do not have a considerable influence on COSMO-CLM variables.





**Figure 4: Mean seasonal cycles of total evapotranspiration and the amount of water evaporation over the (a, b) Parc, (c, d) Linden and (e, f) Lindenberg research domains. Climatological means for CCLMref (blue line), CCLMv3.5 (green line), CCLMv4.5 (brown line), CCLMv4.5e (red line), GLEAM_v3.5a (dotted line) and GLEAM_v3.5b (dashed line) are calculated for the period 2010–2015**

The first visible changes begin to appear in April. These changes are related to increasing air temperature and beginning of

plant growth. The maximum changes between experiments occur from June to August, when vegetation has the greatest





influence on evapotranspiration. We assume that the original version of COSMO-CLM model underestimates the values of ZVERBO and AEVAP parameters due to the lower limit of stomatal resistance (in-situ data can be fewer than the lower limit of COSMO-CLM). The third period begins from September to October. This period is described by leaves wilting and progressive reduction of differences between the experiments. The statistical parameter RMSE for ZVERBO and AEVAP

variables (Fig. 5a) confirmed the visual assessment.

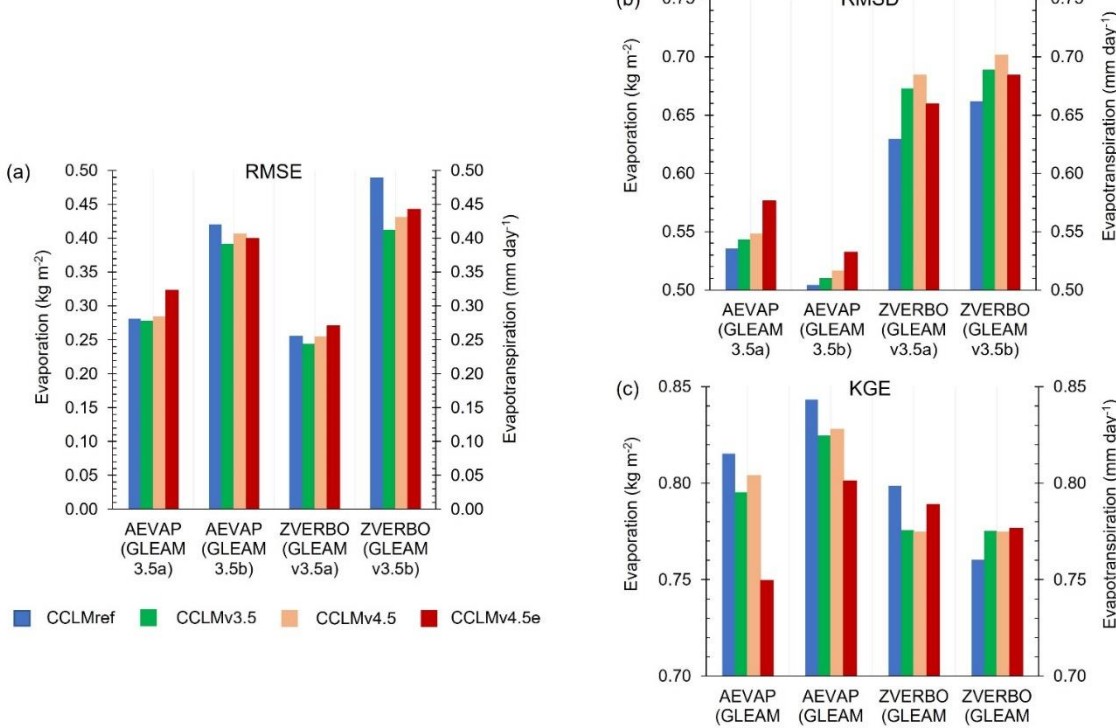

**Figure 5: Model performance for evaporation and evapotranspiration over the Parc domain for the different model experiments: CCLMref (blue), CCLMv3.5 (green), CCLMv4.5 (brown) and CCLMv4.5e (red). The considered scores are: (a) the RMSE calculated from the differences (model experiment minus observation) taken at sites, (b) the RMSD and (c) the KGE taken at each**

**grid cell for each day (daily means) over the period 2010–2015**

This comparison with GLEAM datasets (v3.5a and v3.5b) demonstrates that canopy processes are realistically represented in the new algorithms and the results are slightly better in comparison with the control experiment. The comparison results of the experiments for stations (by 3 domains) show that calculations are similar, but the experimental data are slightly better than the original COSMO-CLM_v5.16. In compare with the GLEAM datasets, the experiments CCLMv3.5 ($RMSE_{ZVERBO}$ = 0.328,

$PCC_{ZVERBO}$ = 0.886 and $RMSE_{AEVAP}$ = 0.334 $PCC_{AEVAP}$ = 0.892) and CCLMv4.5 ($RMSE_{ZVERBO}$ = 0.343, $PCC_{ZVERBO}$ = 0.882 and $RMSE_{AEVAP}$ = 0.345 $PCC_{AEVAP}$ = 0.888) have demonstrated better results than the reference experiment – CCLMref ($RMSE_{ZVERBO}$ = 0.372, $PCC_{ZVERBO}$ = 0.875 and $RMSE_{AEVAP}$ = 0.351, $PCC_{AEVAP}$ = 0.881). We have also compared these parameters (presented at COSMO-CLM grid) with GLEAM datasets (interpolated to COSMO-CLM grid) based on the statistical parameters (RMSD – Fig. 5b, PCC, DAV, KGE – Fig. 5c). The KGE values for AEVAP and ZVERBO parameters



are higher than 0.74 for all simulations. The highest performance values for GLEAM datasets are obtained with the simulation
based on the experiment – CCLMref ($KGE_{AEVAP} = 0.829$, $KGE_{ZVERBO} = 0.780$). The experiment CCLMv4.5 demonstrated the
best scores from new updates ($KGE_{AEVAP} = 0.816$, $KGE_{ZVERBO} = 0.775$). The experiments CCLMv3.5 ($KGE_{AEVAP} = 0.810$,
$KGE_{ZVERBO} = 0.776$) and CCLMv4.5e ($KGE_{AEVAP} = 0.776$, $KGE_{ZVERBO} = 0.783$) showed better results for ZVERBO parameter
than the experiment CCLMv4.5, however the accuracy of AEVAP parameter is less than CCLMv4.5. The RMSD statistical

parameter for the data presented at COSMO-CLM grid has also confirmed that the reference experiment has the lowest values
of errors for these parameters ($RMSD_{AEVAP} = 0.520$, $RMSD_{ZVERBO} = 0.646$). In our updates, CCLMv3.5 has better results for
AEVAP parameter ($RMSD_{AEVAP} = 0.527$), and CCLMv4.5e for ZVERBO ($RMSD_{ZVERBO} = 0.672$). The spatial correlation
coefficients of the simulations with the GLEAM datasets for AEVAP and ZVERBO are similar and equal to 87%. The all-
experiment simulations for ZVERBO parameter have an improvement in performance indicated by the positive DAV values.

The experiments for AEVAP parameter have a similar to CCLMref or less of performance indicated by the DAV values equal
to zero or negative values. Analogically data are obtained for AEVAP and ZVERBO variables for GLEAM v3.5b dataset.

### 4.3 Sensible and latent heat fluxes

Sensible (model variable – ASHFL) and latent (model variable – ALHFL) heat fluxes are the key elements of the equation of
radiation balance and play an important role in the heat and water vapor transfer (de Noblet-Ducoudre and Pitman, 2021). It

was important to look at the changes in these fluxes due to changes in stomatal resistance algorithms. In order to validate the
experiments, we applied the information from the EURONET database (for parc domain) and the data from Linden (linden
domain) and Lindenberg (lindenberg domain) sites. The comparison of the model data which are presented on the COSMO-
CLM grid (2.2 km distance between grid points) with the in-situ data demonstrates that the experiments have significant
differences with the in-situ data. However, we expected these results, and the similar situation was described in (Osborn and

Hulme, 1998).  The experiment results over three study regions for ASHFL and ALHFL model parameters are shown in (Fig.
6a, 6b) as climatological annual cycles averaged over 2010 to 2015 years. The results show that the experiment results are
similar to the CCLMref data. Statistical analysis for latent and sensible heat fluxes shows that the experiments CCLMv3.5
($RMSE_{ASHFL} = 26.05$, $PCC_{ZVERBO} = 0.678$ and $RMSE_{ALHFL} = 28.25$ $PCC_{AEVAP} = 0.639$) and CCLMv4.5 ($RMSE_{ASHFL} = 26.01$,
$PCC_{ASHFL} = 0.676$ and $RMSE_{ALHFL} = 28.36$ $PCC_{ALHFL} = 0.635$) have demonstrated better results than the CCLMref

($RMSE_{ASHFL} = 26.14$, $PCC_{ASHFL} = 0.674$ and $RMSE_{ALHFL} = 28.34$ $PCC_{ALHFL} = 0.635$) and CCLMv4.5e ($RMSE_{ASHFL} = 27.36$,
$PCC_{ASHFL} = 0.675$ and $RMSE_{ALHFL} = 28.85$, $PCC_{ALHFL} = 0.624$). The field data are not compared to each other because we
didn't use projects with in-situ latent and sensible heat fluxes which are presented on grid.





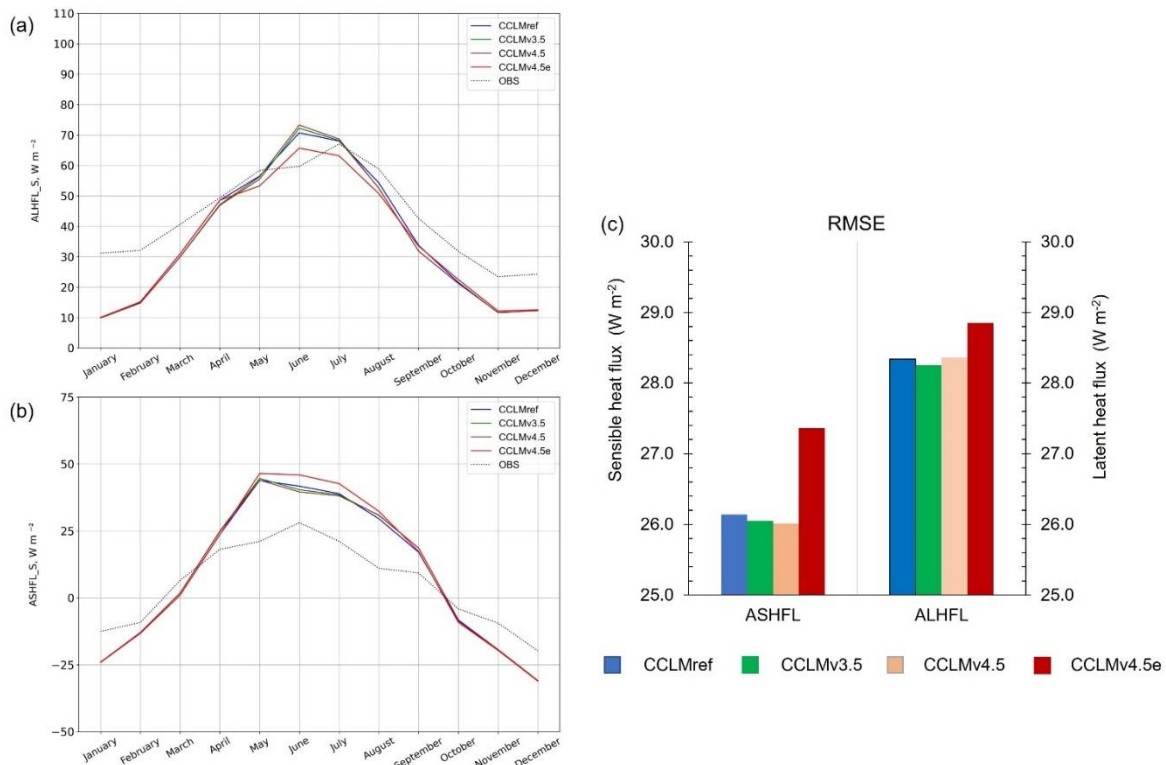

**Figure 6: Mean seasonal cycle of latent (a) and sensible (b) heat fluxes and model performance (c) over the Parc domain.**
**Climatological means for CCLMref (blue line), CCLMv3.5 (green line), CCLMv4.5 (brown line), CCLMv4.5e (red line) and EURONET (dotted line) are calculated for the period 2010–2015. The considered score is the RMSE calculated from the differences (model experiment minus observation) taken at sites over the same period.**

### 4.4 Temperatures

The (Fig. 7) displays differences between experiments CCLMref, CCLMv3.5, CCLMv4.5 and CCLMv4.5e for surface
(Fig. 7a), maximum (Fig. 7b) and minimum (Fig. 7c) temperatures as climatological annual cycles averaged over 2010 to 2015 years. Before the validation, we assumed that the changes in temperature parameters should be minor and reflect the changes in evaporative fraction with less stomatal resistance data leading to enhanced evapotranspiration and consequently reduced sensible heating thus lowering temperatures (surface, maximum, minimum) (Tölle et al. 2014, Tölle and Churiulin 2021). The validation and statistical analysis of these parameters confirmed our assumption. The results of the statistical analysis (for
stations) based on RMSE for temperatures are presented in (Fig. 7d). The experiments CCLMv3.5 ($RMSE_{TS, Tmax, Tmin}$ = 1.749, 10.04, 7.648) and CCLMv4.5 ($RMSE_{TS, Tmax, Tmin}$ = 1.724, 10.05, 7.599) have demonstrated better results than the CCLMref ($RMSE_{TS, Tmax, Tmin}$ = 1.725, 11.23, 9.673) and CCLMv4.5e ($RMSE_{TS, Tmax, Tmin}$ = 1.838, 10.06, 7.677). The Pearson correlation coefficient for all simulations is above 0.98 for surface, 0.93 for maximum and 0.92 for minimum temperatures.

The statistical analysis of the field data also demonstrates that the differences between the temperature simulations are small (Fig. 7e and 7f). Still, consistent performance discrepancies can be extracted. The performance values based on KGE for $T_S$



are higher than 0.90 for all experiments. The highest performance value is 0.914 based on the CCLMv4.5 and CCLMref ($KGE_{TS}$ = 0.913) experiments. The third highest performance value of 0.912 is found by the simulation based on CCLMv3.5. The KGE of the simulation based on CCLMv4.5e is the lowest with a value of 0.907. The RMSE for $T_S$ is the lowest with

1.213 for the simulations based on CCLMref, followed by CCLMv4.5 with 1.217, and then CCLMv3.5 and CCLMv4.5e with 1.227 and 1.253. The spatial correlation coefficients of the simulations with the HYRAS observational dataset are similar and equal to 0.99. The new experiments have a similar (CCLMref) of performance indicated by the DAV values equal to zero. Similar statistical results are extracted for $T_{max}$ and $T_{min}$ climatic variables.








**Figure 7: Mean seasonal cycles of temperatures: surface (a), maximum (b), minimum (c) and model performances over the Parc domain for the different model experiments: CCLMref (blue), CCLMv3.5 (green), CCLMv4.5 (brown), CCLMv4.5e (red) and HYRAS (dotted line). Climatological means and model performances: (d) the RMSE from the differences (experiment minus observation) taken at sites, (e) RMSD and (f) KGE taken at grid points are calculated for the period 2010–2015.**




It should be noted that the experimental stomatal resistance data demonstrate a positive trend towards increasing precision of this COSMO-CLM parameter. Moreover, the results show that changes in stomatal resistance and photosynthesis algorithms can improve the accuracy of other variables of COSMO-CLM model (e.q.: AEVAP, ZVERBO, ALHFL, ASHFL), which in turn demonstrates the need for further development, including plant growth and phenology changes. We have assumptions

similar to the reference experiment assumptions (Doms et al., 2018) which are related to the TERRA-ML characteristics. Because of that there is a chance that the accuracy of stomatal resistance calculations can be additionally improved through the changes in these assumptions.

The difference in results between statistical analysis in one point (station) and statistical analysis of the data presented on

COSMO-CLM grid is related to the different plant types which are presented in the research domains. We have tried to find the research domains with the vegetation, which is presented only $C_3$ grass, because only the two PFT ($C_3$ and $C_4$ grass) were implemented in COSMO-CLM instead of the appropriate land use classes. Nevertheless, this task was truly challenging and there is the vegetation in the research domains which is presented not only by $C_3$ grass (for example: croplands). In that case, the COSMO-CLM model which applies the corresponding land use classes (Doms et. al, 2018) demonstrates better results.

**5 Conclusions**

Evapotranspiration plays an important role in determining the component of energy balance. One of the variables belonging to evapotranspiration is stomatal resistance. Stomatal resistance allows to evaluate plant physiological response to dynamic biophysical, environmental, soil water conditions and $CO_2$ concentration of the immediate surrounding of the leaf (Übel, 2015). Nevertheless, COSMO-CLM uses simplified phenology schemes, which are in general not capable of modelling complex

processes depending on day length, temperature, and water availability relevant to the start of the growing season in spring, the evolution of the leaf area index and plant coverage and the senescence in autumn. Because of that, in COSMO-CLM model we implemented the stomatal resistance and leaf photosynthesis algorithms from CLM models where the stomata resistance depends on of physical, biophysical, and biogeochemical processes that simulate the terrestrial radiation, heat, water and carbon fluxes in response to climatic forcing (Stöckli et al., 2008).


The three new versions of COSMO-CLM v5.16 have been developed in the course of the research. The first version (CCLMv3.5) applies the stomatal resistance and leaf photosynthesis algorithms based on CLM 3.5. The second CCLMv4.5 is based on the phenology algorithms of CLM v4.5 including the soil water stress function. The last one - CCLMv4.5e - is similar to the CCLMv4.5, but with adapted equations for dry leaf calculations. The new versions of COSMO-CLM take into account

the difference of the physiological properties between sunlit and shaded leaves. New algorithms use the modern physically based approach to stomatal resistance. The prognostic environmental parameters for calculations of stomatal resistance do not depend on the stomatal resistance values (as it was before) and are connected with each other by leaf photosynthesis. Stomatal





resistance values are influenced by atmospheric $CO_2$ concentration and the leaf photosynthesis and $CO_2$ uptake can be calculated.


The new versions were compared with the in-situ and the reference experiment (CCLMref) data. The experiments CCLMv3.5 and CCLMv4.5, in most cases demonstrate better results than reference experiment in comparison with the in-situ data for the research domains over Germany. The greatest differences between these experiments and CCLMref were fixed for stomatal resistance. When compared with the real data the new experiments results prove to have lower values of stomatal resistance.

The less values of stomatal resistance are better suited to the in-situ data. The changes in stomatal resistance have a visible positive reaction to the changes in evapotranspiration and evaporation. For our research domains, we found that the less values of stomatal resistance (all experiments) lead to increased values of evaporation and transpiration. The comparison of evaporation and evapotranspiration modelling results with GLEAM dataset show that the new experiments have better correlation and less values of RMSE then control experiment. We have also noted that the enhanced evapotranspiration values

lead to consequently reduced sensible heating (except experiment CCLMv4.5e), resulting in a decrease in temperatures. Moreover, the changes in maximum temperature were noted: for the "Parc domain" the values of maximum temperature increased, while the values of $T_{max}$ for Linden and Lindenberg domains remain unchanged. The results indicate that changes in stomatal resistance and photosynthesis algorithms can improve the accuracy of other parameters of the COSMO-CLM model by comparing them with EURONET, HYRAS, GLEAM dataset and meteorological observations at the sites.


Nevertheless, we didn't change the phenological cycle of COSMO-CLM (yet), which is still based on a 6-year climatology cycles and follows the same sinusoidal fitted curve between its maximum and minimum value each year neglecting any influence or feedback on the environmental conditions. Moreover, we have made several assumptions which we applied for the CLM algorithms implementation (1 – calculate values of the atmospheric specific humidity at surface layer instead of the

leaf surface; 2 – apply the temperature of near surface layer instead of the vegetation temperature; 3 – calculate the saturation vapour pressure at near surface temperature ($e_i^{*T_s}$) instead of the saturation vapor pressure inside the leaf). Unfortunately, we cannot correct these assumptions without global changes in COSMO-CLM, however we will try to implement the new algorithms for calculations of phenology (including leaf area index and plant coverage) based on more modern, dynamic algorithms take into account the carbon uptakes rate, changes in temperature and the evolution of the growing season in spring

and the senescence in autumn.

**Appendix A: Stomatal resistance algorithm of CLM model**

The calculation of leaf photosynthesis in CLM3.5 is based on the model of Farquhar for $C_3$ (Farquhar et al. 1980) and Collatz for $C_4$ plants (Collatz et al., 1991). According to the CLM3.5 strategy, the minimum rate set by one of the limitation relations controls $CO_2$ assimilations at the leaf level Eq. (A1).



$A = \min(w_c, w_j, w_e)$, (A1)

where $w_c$ is the rate of $CO_2$ fixation in the carboxylation of RuBP in the Calvin cycle [µmol $CO_2$ m$^{-2}$ s$^{-1}$], $w_j$ is the maximum rate of carboxylation allowed by the capacity to regenerate RuBP [µmol $CO_2$ m$^{-2}$ s$^{-1}$], $w_e$ is the capacity for the export or utilization of the carbohydrates [µmol $CO_2$ m$^{-2}$ s$^{-1}$].

$$w_c = \begin{cases} \dfrac{V_{cmax}(C_i - \Gamma_*)}{C_i + K_c\left(1 + \frac{O_i}{K_o}\right)} & for\ C_3\ plants \\ V_{cmax} & for\ C_4\ plants \end{cases} \Bigg\} \ C_i - \Gamma_* \geq 0,$$ (A2)

$$w_j = \begin{cases} \dfrac{(C_i - \Gamma_*)4.6\phi\alpha}{C_i + 2\Gamma_*} & for\ C_3\ plants \\ 4.6\phi\alpha & for\ C_4\ plants \end{cases} \Bigg\} \ C_i - \Gamma_* \geq 0,$$ (A3)

$$w_e = \begin{cases} 0.5V_{cmax} & for\ C_3\ plants \\ 4000V_{cmax}\dfrac{C_i}{P_{atm}} & for\ C_4\ plants \end{cases} \Bigg\} \ C_i - \Gamma_* \geq 0,$$ (A4)

where $V_{cmax}$ is the maximum rate of carboxylation [µmol $CO_2$ m$^{-2}$ s$^{-1}$], $\Gamma_*$ is the $CO_2$ compensation point [Pa], $C_i$ is the intercellular $CO_2$ pressure [Pa], $K_c$ and $K_o$ are the Michaelis-Menten constants for $CO_2$ and $O_2$ [Pa], $O_i = 0.209P_{atm}$ is the $O_2$ partial pressure, $\phi$ is the absorbed photosynthetically active radiation [W m$^{-2}$], $\alpha$ is the quantum efficiency [µmol $CO_2$ per

µmol photons], 4.6 is coefficient for converting photosynthetic photon flux. It should be noted that RuBisCO limitation ($w_c$) and the upper limit of the capacity utilization ($w_e$) depend on $V_{cmax}$ which is a function of several environmental variables considering for the vertical canopy integration scheme (Thornton and Zimmermann, 2007).

$$V_{cmax} = V_{cmax25}(2.4)^{\frac{T_v - 25}{10}}F_{T_v}\ F_{DYL}\ F_N\ F_{wat},$$ (A5)

where $V_{cmax25}$ is the maximum rate of carboxylation at 25 °C [µmol $CO_2$ m$^{-2}$ s$^{-1}$], $T_v$ – leaf temperature, however in the case

of COSMO-CLM, the surface air temperature was used [K], $F_{T_v}$ is a function that mimics thermal breakdown of metabolic processes Eq. (A6) at temperatures exceeding 35 °C, $F_{DYL}$ is day length function Eq. (A7) with maximum values of $V_{cmax}$ at the maximum day length in summer, $F_N$ is the nitrogen availability factor and $F_{wat}$ is the function which depends on the soil water potential Eq. (A8) in each soil layer (Thornton and Zimmermann, 2007), COSMO-CLM doesn't work with soil water potential parameters therefore we modified this parameter to soil water content for each soil layer, that allowed us to apply Eq.

(B6) for calculation $V_{cmax}$ and save a soil water balance in COSMO-CLM.

$$F_{Tv} = \left[1 + \exp\left(\frac{-220000 + 710(T_v + T_f)}{0.001R_{gas}(T_v + T_f)}\right)\right]^{-1},$$ (A6)

where $T_f = 273.15\ K$ is the freezing temperature of water.

$$F_{DYL} = \frac{DYL^2}{DYL_{max}^2},$$ (A7)



where $DYL$ is daylength [second] and $DYL_{max}$ is a maximum day length.

$$F_{wat} = \sum_k w_i r_i, \qquad (A8)$$

where $w_i$ is a plant wilting factor for layer $i$ and $r_i$ is the fraction of roots in layer $i$. The maximum rate of carboxylation varies with foliage nitrogen concentration and specific leaf area at 25 °C is calculated as Eq. (A9):

$$V_{cmax25} = N_a F_{LNR} F_{NR} \alpha_{R25}, \qquad (A9)$$

where $N_a$ is the area-based leaf nitrogen concentration [gNm$^{-2}$ leaf area] which is defined by (eq. A10), $F_{NR} = 7.16$ is the
mass ratio of total RuBisCO molecular mass to nitrogen in RuBisCO [gRubisco g$^{-1}$N in Rubisco], $F_{LNR}$ is the fraction of leaf nitrogen in RuBisCO [gN in Rubisco g$^{-1}$N], $\alpha_{R25}$ is the specific activity of RuBisCO (µmol CO$^2$ g$^{-1}$ Rubisco s$^{-1}$).

$$N_a = \frac{1}{CN_L\,SLA}, \qquad (A10)$$

where $CN_L$ is the leaf carbon-to-nitrogen ratio [gCg$^{-1}$N]. $SLA$ – specific leaf area index which is calculated separately for sunlit and shaded leaves.

The Eq.(A5) demonstrates the dependency of the leaf photosynthesis rate based on Farquhar or Collatz model on environmental conditions, aggregates them with the Ball-Berry stomatal model (Ball, 1988) and explains how we can use the "two-big leaf" approach for calculating it for sunlit and shaded leaves.

**Code and data availability**

The latest version of scripts which were created and applied for this research are available as a Python package from
https://github.com/EvgenyChur/PT-VAINT (last access: 01 November 2021) under the GPLv3 license. The CLM (3.5) model code in online view is available in open access from the official web-page of CLM community: https://www.cesm.ucar.edu/models/cesm1.2/cesm/cesmBbrowser/html_code/clm/ (last access: 11 May 2021), moreover the CLM model code can be downloaded for your PC from the NCAR web-page: https://www.earthsystemgrid.org/dataset/ucar.cgd.clm.src.3.5.html (last access: 25 March 2021). The COSMO-CLM model
code can be available after the registration for the official members of the COSMO consortium https://wiki.coast.hzg.de/clmcom (last access: 27 April 2021). The validation data from Lindenberg was provided for request by Claudia Becker (Claudia.Becker@dwd.de), from Linden the data was obtained from the official web-page system of requests (http://www.meteocentrale.ch/en/info/contact.html, last access: 10 January 2021). The validation datasets from EURONET are available upon request from European Fluxes Database Cluster http://www.europe-fluxdata.eu/ (last access: 5
April 2021), E-OBS https://www.ecad.eu/download/ensembles/download.php (last access: 09 April 2021), HYRAS https://www.dwd.de/DE/leistungen/hyras/hyras.html (last access: 09 April 2021) and GLEAM https://www.gleam.eu/ (last access: 09 April 2021).



**Author contributions**

Conceptualization, project administration and supervision: MT. Investigation and methodology: MT, JH, J-MB, VK, MU, US, EC. Data curation: MT, EC. Validation: MT, JH, J-MB, VK, MU, EV. Writing – original draft: EC. Writing, review and editing: EC, MT. All authors reviewed and approved the final version of the paper.

**Competing interests**

The authors declare that they have no conflict of interest.

**Acknowledgements**

Thanks is due to Rüdiger Schaldach for reviewing this manuscript. We thank Claudia Becker for her assistance at the Lindenberg site, Polina Govorina for her help with English language. Finally, we thank the reviewers and the editor, who significantly contributed to the improvement of the paper.

**Financial support**

This research was funded by the German Research Foundation (DFG) through grant number 401857120.

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
