# Peer review of "Improving the stomatal resistance, photosynthesis and two big leaf algorithms for grass in the regional climate model COSMO-CLM"

_Biogeosciences, 2021_

## Community Comment (CC2)

L.A. Timofeeva
tilarisa@gmail.com

The international group of authors shares the findings of their research on improving several algorithms included in COSMO-CLM. They have applied the model as a flexible tool for taking into account land-atmosphere fluxes and phenological properties, which can change due to global warming. The authors' idea is new (at least to me). The interdisciplinary core of the research makes it especially worth-while and up-to-date.

However, to help readers to understand quite complex content the authors have to pay more attention to details and relevant explanation of principal matters. In my opinion, the following terms need clarification in Abstract to attract more readers, who are not experts in plants.

**C3 vs. C4 plants** (first met in Line 13)
It would be useful to address the main differences between the two types of plants, which are due to their ability and manner to fix carbon dioxide in cool and warm seasons. Please provide some examples of "$C_3$ grass plants in Germany" just to make the picture more vivid and realistic.

**One-big leaf vs. two-big leaf approach** (first met in Lines 14-15)
The difference between these two approaches is critical. Implementation of a two-big leaf approach means dividing vegetation into sunlit and shaded portions. This has complicated calculating dramatically and is worth highlighting, even though the approach is explained in subsection 2.4.2.

**Dry leaf calculations** (first met in Line 21)
What parameter do the authors imply? Please explain. I could not find either definition or equations in the paper.

There are other points in the paper that need to be improved to enhance its readability.

Line 10. "Climatic changes towards warmer temperatures" can be replaced with a more common term "Global warming" that looks suitable in the context.

Line 65. The full definition of LAI (the Leaf area index) should be placed before the abbreviation.

Line 95. Equations 7 and 8 look the same. Please check.

Line 264. "The focus of the research was on statistical analysis of the summer months" has no sense, since not months but certain parameters can be analyzed, which are listed in the next sentence.

Lines 303-304. There is a statement that the absolute error for precipitation is 4.45%. It must be corrected, since the absolute error cannot be in %.

Line 357. It is not clear where in North America is situated the region for which the published in-situ stomatal resistance data were found. This is important to decide if the data can be used for validation of the results.

Line 372. Table 3. I find myself doubtful what CTR stands for. Besides, the word DATE seems strange in two columns between v3.5 and v4.5e.

Line 486. "Evapotranspiration plays an important role in determining the component of energy balance". What component do the authors mean?

Moreover, I have noted a number of issues that need rewriting. In my opinion, too many punctuation marks are used, such as dash, colon, and brackets. Besides, spelling of the world "parameterization" and use of articles are not consistent.

I should confess English is not my native tongue; moreover, the authors' vocabulary and writing style might differ from mine. However, some English language flaws are obvious and should be corrected. I consider the shortcomings noted by me might be consequences of the authors' urgent willing to share their latest findings.

I have also paid attention to the fact that several references, explaining basic foundations, were made to modern resources instead of appropriate older ones. One example is in Line 34, de Noblet-Ducoudre and Pitman, 2021. There is classic research on the role of soil and its parameters in land-atmosphere interaction, starting from Dokuchaev. All of all, the Reference list includes a wide range of resources, which are accessible via the links provided. This is only one of the numerous strengths the paper has.

To begin with, the paper is perfectly structured. Each section logically presents all the information announced in Abstract. In section 2, the changes implemented in the TERRA-ML and COSMO-CLM models are described. These noticeable changes have made applying COSMO even more time and resource consuming than it usually is. The modified algorithms are quite detailed described in the text and in Appendix A. Codes as a Python package and data are available via the links provided. All this proves that the authors are experienced data scientists and programmers.

Thorough statistical analysis of a number of modeled characteristics was carried out based on the relevant parameters, including the Kling-Gupta Efficiency index, which has become quite popular recently. The choice of different validation datasets seems to be convincing. What I miss is a more detailed discussion in terms of botanic.

I would like to highlight that the authors reasonably evaluate perspectives of possible future research; they will try to implement new algorithms, which will enable accounting carbon uptake rate, changes in temperature and different growing seasons.

---

## Author Comment (AC1)

L.A. Timofeeva
tilarisa@gmail.com

The international group of authors shares the findings of their research on improving several algorithms included in COSMO-CLM. They have applied the model as a flexible tool for taking into account land-atmosphere fluxes and phenological properties, which can change due to global warming. The authors' idea is new (at least to me). The interdisciplinary core of the research makes it especially worth-while and up-to-date.

However, to help readers to understand quite complex content the authors have to pay more attention to details and relevant explanation of principal matters. In my opinion, the following terms need clarification in Abstract to attract more readers, who are not experts in plants.

**C3 vs. C4 plants** (first met in Line 13)
It would be useful to address the main differences between the two types of plants, which are due to their ability and manner to fix carbon dioxide in cool and warm seasons. Please provide some examples of "C3 grass plants in Germany" just to make the picture more vivid and realistic.

*Answer:* Ok, we agree with the reviewer. The sentence with the differences between C3 and C4 plants was added to the manuscript.

**One-big leaf vs. two-big leaf approach** (first met in Lines 14-15)
The difference between these two approaches is critical. Implementation of a two-big leaf approach means dividing vegetation into sunlit and shaded portions. This has complicated calculating dramatically and is worth highlighting, even though the approach is explained in subsection 2.4.2.

*Answer:* Thank you for this comment. Yes indeed, the modernization required a lot of additional work for the implementation of this approach. However, it is a programming and technical work and more information about the modernization of the model code can be found in the GitHub repository, appendix A and in the official documentation for COSMO and CLM models. Also, we are working on the special documentation were all these changes will be displayed. Nevertheless, additional changes (corrections) were added to the text. We hope, these changes help to explain the new implementations in more detail.

**Dry leaf calculations** (first met in Line 21)
What parameter do the authors imply? Please explain. I could not find either definition or equations in the paper.

*Answer:* Ok, we agree. The text of the manuscript was corrected. The name of this parameter was changed from *dry leaf calculations* to *the calculations of transpiration from dry leaf surface*. The detailed answer to this question (with equations and more detailed description) is presented in our answer to Sibylle Schaphoff questions and comments (https://doi.org/10.5194/bg-2021-294-CC1).

There are other points in the paper that need to be improved to enhance its readability.

**Line 10.** "Climatic changes towards warmer temperatures" can be replaced with a more common term "Global warming" that looks suitable in the context.

*Answer:* Ok, we agree. The text was corrected. *The new sentence is:* The simplified vegetation algorithm of the regional climate model COSMO-CLM is not capable of modelling complex processes depending on dynamic biophysical, environmental, soil water conditions and CO2 concentrations especially relevant in the context of global warming.

**Line 65.** The full definition of LAI (the Leaf area index) should be placed before the abbreviation.

*Answer:* Ok, we agree. The text was corrected. *The new sentence is*: Moreover, the leaf area index (LAI) does not respond to water stress and depends on vegetation parameters.

**Line 95.** Equations 7 and 8 look the same. Please check.

*Answer:* Thank you for this clarification. The equations were corrected.

**Line 264.** "The focus of the research was on statistical analysis of the summer months" has no sense, since not months but certain parameters can be analysed, which are listed in the next sentence.

*Answer:* Ok, we corrected the text of the manuscript according to this comment. *The new sentence is*: The focus of the research was on validating values of the output COSMO-CLM parameters (near-surface temperature, latent and sensible heat fluxes, fluxes of water evaporation and transpiration, and stomatal resistance) during the period when the vegetation is in active phase.

**Lines 303-304.** There is a statement that the absolute error for precipitation is 4.45%. It must be corrected, since the absolute error cannot be in %.

*Answer:* Ok, thank you for noticing this mistake. The unit of the error was corrected from % to mm.

**Line 357.** It is not clear where in North America is situated the region for which the published in-situ stomatal resistance data were found. This is important to decide if the data can be used for validation of the results.

*Answer:* Ok, we added the information to the manuscript. The region is situated similar climate conditions and the data can thus be used for our analysis.

**Line 372.** Table 3. I find myself doubtful what CTR stands for. Besides, the word DATE seems strange in two columns between v3.5 and v4.5e.

*Answer:* We thank you for this comment. The header of the table 3.1 was corrected. The word DATE was deleted, the new header is v4.5.

**Line 486.** "Evapotranspiration plays an important role in determining the component of energy balance". What component do the authors mean?

*Answer:* Ok, we agree. The sentence is corrected. The new one is: Evapotranspiration is an important component in the energy balance equation.

Moreover, I have noted a number of issues that need rewriting. In my opinion, too many punctuation marks are used, such as dash, colon, and brackets. Besides, spelling of the world "parameterization" and use of articles are not consistent. I should confess English is not my native tongue; moreover, the authors' vocabulary and writing style might differ from mine. However, some English language flaws

are obvious and should be corrected. I consider the shortcomings noted by me might be consequences of the authors' urgent willing to share their latest findings.

*Answer:* Ok, thank you for these comments. We agree with them. The text of the manuscript is adjusted to the comments. We also send our manuscript to a professional editing service.

I have also paid attention to the fact that several references, explaining basic foundations, were made to modern resources instead of appropriate older ones. One example is in Line 34, de Noblet-Ducoudre and Pitman, 2021. There is classic research on the role of soil and its parameters in land-atmosphere interaction, starting from Dokuchaev. All of all, the Reference list includes a wide range of resources, which are accessible via the links provided. This is only one of the numerous strengths the paper has.

*Answer:* Ok, thank you for this information, it is a good idea. We added this classical research paper to our manuscript.

To begin with, the paper is perfectly structured. Each section logically presents all the information announced in Abstract. In section 2, the changes implemented in the TERRA-ML and COSMO-CLM models are described. These noticeable changes have made applying COSMO even more time and resource consuming than it usually is. The modified algorithms are quite detailed described in the text and in Appendix A. Codes as a Python package and data are available via the links provided. All this proves that the authors are experienced data scientists and programmers.

*Answer:* Thank you for this comment. We appreciate it.

Thorough statistical analysis of a number of modelled characteristics was carried out based on the relevant parameters, including the Kling-Gupta Efficiency index, which has become quite popular recently. The choice of different validation datasets seems to be convincing.
Ok, we thank the reviewer for this comment.

What I miss is a more detailed discussion in terms of botanic.
Ok, the reviewer raises a good point. We added more discussion in terms of botanic. For example, in the introduction was added information about C3 and C4 grass. The methods were expanded by the more detailed information about the vegetation parameters. Stomatal resistance section: the new sentence about the role of Vc, max in Ball-Berry approach added to the manuscript.

I would like to highlight that the authors reasonably evaluate perspectives of possible future research; they will try to implement new algorithms, which will enable accounting carbon uptake rate, changes in temperature and different growing seasons.

We thank the reviewer for this thoughtful comment.

---

## Author Comment (AC2)

Our response to Sibyll Schaphoff comments on 04.01.2022 (https://doi.org/10.5194/bg-2021-294-CC1)

*Title of the manuscript:* Improving the stomatal resistance, photosynthesis and two big leaf algorithms for grass in the regional climate model COSMO-CLM

**General:**
The present study compares three different approaches to simulate stomata resistance and the connected against a very simplified approach in the regional climate model COSMO-CLM, which is not capable to simulate vegetation processes dynamically. These processes are very important in the coupling with the atmosphere and thus very important to calculate in more dynamic way.
I extremely appreciate the comprehensive description of the methods, but I think the evaluation needs a broader application for additional variables and sites to better assess the different methods. Furthermore, I encourage the authors to introduce at least one tree as well to evaluate, if these three approaches lead to a better representation of the biosphere-atmosphere interaction.

*Answer:* We thank the reviewer for this comment. We agree that this study needs to be extended for different plants and other study sites. This requires more extensive updates and technical development for the regional climate model and is not our focus here. Thus, it would exploit our current study. However, it will be the focus of our future studies. In the current manuscript, we evaluate evapotranspiration due to the different approaches to simulate stomata resistance coupled to photosynthesis adapted from the Community Land Model over three sites mainly dominated by grass with one-column regional climate model simulations. This is entirely new for COSMO-CLM and need careful evaluation step-by-step. We find that we also need to update our model to time varying leaf area index and plant coverage. After that, we will update the code for different plant functional types and perform simulations over different sites and bigger domains if the results confirm to be more realistic. Therefore, we write in the conclusion section of the manuscript that the next step will be to add time-varying leaf area index and plant coverage. Depending on the results, the model will be extended for different plant functional types and perform simulations over larger domains.

**Detailed comments:**
**Page 4 line 124:** Is T_r and Tr_k the same? Would you please provide how foliage resistance and stomatal resistance is related and if that has changed?

*Answer:* Ok, the reviewer raises a good point. We corrected the text. Here, $T_r$ is the same parameter as $T_{rk}$ .. In accordance with the official documentation of COSMO-CLM, model stomatal resistance is a part of foliage resistance. In particular:

$$T_r = f_{plnt} * (1 - f_i) * (1 - f_{snow}) * E_{pot}(T_{sfc}) r_a (r_a + r_f)^{-1}$$

where: $r_f$ – is foliage resistance, which is equal to:

$$r_f^{-1} = r' C_F$$

where: $r'$ describes the reduction of transpiration by the stomatal resistance.

$$r' = r_{la}(r_{la} + r_s)^{-1}$$

where: $r_s$ – is stomatal resistance and $r_{la}$ is

$$r_{la}^{-1} = C' u_*^{0.5}$$

where: $u_*$ is frictional velocity.

**Page 5 line 150:** A new description of a new parametrization scheme for the maximum rate of carboxylation is mentioned, please give the link to appendix, and explain Eq. A8 in more detail. What is meant with the plant wilting factor and what is k?

*Answer:* Ok, we agree with the reviewer and give more explanations. The new text is: The new description of the stomatal resistance in TERRA-ML is calculated on the basis of the plant physiological

approach (Ball et al, 1987; Ball, 1988) with algorithms for canopy fluxes based on Collatz model (Collatz et al., 1991) and improved by (Thornton and Zimmermann, 2007) through the implementation of a new parameterization scheme for the maximum rate of carboxylation ($Vc,max$), which is presented in (Eq. A5) and was the most critical problem of the Collatz model. Also, the parameter $k$ (soil layers) (Eq. A8) was changed to $i$. The equation was corrected. The plant wilting point ($w_i$) is the available water in the i$^{th}$ soil layer relative to an optimal water content. In our research we used the available COSMO-CLM parameter [Doms et.al., 2018] which calculated based on the equation

$$w_i = \frac{\omega_{l,root} - \omega_{PWP}}{\omega_{TLP} - \omega_{PWP}}$$

where $\omega_{l,root}$ is the water content of the soil averaged over the root depth, $\omega_{PWP}$ is the permanent wilting point, $\omega_{TLP}$ is the turgor loss point.

**Page 5 line 170:** If the former version does not calculate photosynthesis, could you give a brief overview, how plants are represented in the model?

*Answer:* Ok, thank you for this comment. The section was corrected. Plants are represented in the COSMO-CLM model by the following vegetation parameters, which are read in by the model as external 2D fields coming from remote sensing data. The vegetation parameters, which are read in, are leaf area index, plant coverage, root depth and roughness length.

**Page 6 eq.4:** Why is only the minimum stomatal conductance influenced by the soil water stress function. Please give the equation of this function. Why is parameter b so different in the two different calculations?

*Answer:* We thank the reviewer for this comment. The values of stomatal resistance (conductance) depend on the several parameters including: daylengths, temperature, photosynthetic active radiation, soil water stress and $CO_2$ concentration. These parameters are included in the algorithm for calculation of $V_{c,max}$. The values of $V_{c,max}$ is used for calculations of leaf photosynthesis ($A_n$) and then in stomatal conductance. It is common part for both versions of the Community Land Model, which we adapted, and works for calculations of stomatal resistance when $A_n > 0$. Then, there are differences in calculation of night values ($A_n = 0$) of stomatal conductance in two versions on CLM model. In CLM 3.5 the algorithm for calculation of stomatal conductance is:

$$g_{st} = \frac{1}{r_s} = m \frac{A_n}{c_s} \frac{e_s}{e_i^*} P_{atm} + b,$$

The night values of stomatal conductance are equal to $b$, because the right part of the equation is equal to zero. The $b$ is empirical parameter, responsible for minimum stomatal conductance. In CLM 4.5 the algorithm for calculation of stomatal conductance is:

$$g_{st} = \frac{1}{r_s} = m \frac{A_n}{c_s} \frac{e_s}{e_i^*} P_{atm} + b\beta_t,$$

It means that night values are depending on soil water stress. The values of parameter $b$, which we used, were from the official documentation of CLM model (version 3.5 and 4.5). We assume that the differences between values in min. stomatal conductance is related to the soil water stress function. For example: if we have values of soil stress function equal to 0.2 the values of min. stomatal conductance from version 3.5 and 4.5 will be equal to each other. We refer to the documentation of the Community Land Model for Version 3.5 and 4.5.

**Page 7 Eq. 7 and 8** are identical.

*Answer:* Thank you for this remark. The equations are corrected.

**Page 8 line 237:** Could you explain more precisely what is meant with "adapted equations for dry leaf calculation". Best would be to add a link to the equation that is used. For the other experiments I'm

maybe able to identify the differences, but an overview table would definitely help to understand these differences much easier.

*Answer:* Thank you for these comments. In COSMO-CLM model there is a parameter (ztraleav – transpiration rate of dry leaves). In the original version of COSMO this parameter is calculated based on this equation:

$$ztraleav = \frac{(zep_s * tai)}{(sai + (zrla + zrstom) * zcatm)}$$

Where: $zep_s$ is potential evaporation, $tai$ is transpiration area index (external data), $sai$ is surface area index (external data), $zrla$ is atmospheric resistance, $zrstom$ is stomatal resistance, $zcatm$ is transfer function. In the Community Land Model there is parameter vegetation transpiration (***qflx_tran_veg***) which is related to the potential evaporation through transpiration (***rppdry***) from leaf and potential latent energy flux (***efpot***):

$$qflx\_tran\_veg = efpot * rppdry$$

$$efpot = forc_{rho} * wtl * (q_{satl} - q_{af})$$

Where: $forc_{rho}$ is air density, $wtl$ is heat conductance for leaf, $q_{satl} - q_{af}$ is humidity gradient

$$rppdry = fdry * rb * \frac{\dfrac{L^{sun}}{rb + r_{s,sun}} * \dfrac{L^{sha}}{rb + r_{s,sha}}}{elai}$$

Where: $fdry$ is fraction of foliage that is green and dry, $rb$ is boundary layer resistance, $r_{s,sun}$ and $r_{s,sha}$ are stomatal resistance of sunlit and shaded leaves, $elai$ is one-sided leaf area index with burying by snow. In our research (experiment v4.5 e) we adapted this algorithm to COSMO-CLM and change the equation for $ztraleav$ on $qflx\_tran\_veg$.

**Page 9 Table:** Do you mean v4.5 instead of Date in table header?
*Answer:* Thank you for this remark. The header of the table was corrected *Date* changed to *v4.5*

Comparison of the stomatal resistance shows that all versions seem to be too high for all regions. Do you have a reason not to adjust the parameter values? Have you any other indication that would disagree with lower stomatal resistance values?

*Answer:* Thank you for this comment. Yes, all stomatal resistance data seem to be high for all regions and we acknowledge this in our manuscript and write that further sensitivity tests on parameter values would be necessary. However, in the current manuscript we used values which were given and officially published.

I would appreciated a comparison to Vmax (leaf photosynthesis carboxylation capacity values) which are very common and available from the TRY database (https://www.try-db.org/TryWeb/dp.php) as well as stomata conductance. That would make the evaluation more valuable and would demonstrate that the models are able to represent the relation between Vmax and stomatal conductance well, and as the manuscript emphasizes the coupling between photosynthesis and transpiration.

*Answer:* Thank you for this information. We downloaded data and found one appropriate dataset, which can be used for our purposes (timeseries with coordinates and data for C3 grass). The new data will be added to the plots.

Please add some statistical values to the evaluation plots against observational data that always helps to assess the results. That's what you did in figure 5, but you could also just add that to the legend on the

plot than it is available at a glance. Is figure 5 done for the three domains only? It's not indicated in the caption, but I assume it.

*Answer:* Thank you for this comment. We added more statistical values to the evaluation plots. According to the figure 5. This figure was created only for one research domain Parc.

You conclude that the implementation would be valuable for the regional climate model, could you indicate which approach you are going to introduce.

*Answer:* We think that the experiment CCLMv4.5 show the better results. The experiment based on the adapted algorithm form CLM version v4.5 is most perspective. This version has an updated algorithm for stomatal resistance and leaf photosynthesis based on the more modern version of CLM model. This algorithm has a regulation function of night values of stomatal resistance depending on soil water stress function and the statistical results for this experiment has slightly better values that other experiments and original version of COSMO-CLM. At the same time, the experiment CCLMv4.5e has more inaccuracies caused by the implementation of the new parameters for calculating transpiration from dry leaf surface which were partly unavailable in the original version of COSMO-CLM and was changed to the constant parameter or COSMO-CLM analogies. The version CCLMv4.5e required additional work for adaptation and further validation of COSMO-CLM parameters.

---

## Author Comment (AC4)

Our response to Referee 1 comments on 31.01.2022 (https://doi.org/10.5194/bg-2021-294-RC1)

*Title of the manuscript:* Improving the stomatal resistance, photosynthesis and two big leaf algorithms for grass in the regional climate model COSMO-CLM

Dear authors,

The subject of the manuscript is for sure very relevant for the climate modelling community dealing with land-surface processes and their interactions with the atmosphere. However, I'm sorry to conclude that this manuscript does not fulfil the expectations I have on a scientific documentation of theory, experiments and results. The reason is partly that the English language is now at such a level where it becomes difficult to actually understand what the authors wish to describe in some background and results.

*Answer:* Ok, thank you for these comments. We corrected the English language and sent the manuscript to an English editing service. We hope that the manuscript is now clear it its meaning and the English in a good state. Please see the revised manuscript.

The reasons are also that the structure, motivation and balance of the text and results are not satisfactory. For example, the Introduction section lacks clarity (see below), Section 2.5 "Statistical analysis" refers to details that are described in later sections. It is not clear what is the objective with Section 4 "Results and discussion".

*Answer:* Ok, thank you for pointing this out. We restructured our manuscript for a better readability. We clearly state the motivation of our study in the "Introduction" section now. Moreover, we added current problems of COSMO-CLM and the purposes of our research to the "Introduction" section. We checked the manuscript for a better balance and moved content to the appendix if the balance is exceeded. In particularly, a part of the statistical approach in the "Methods" section is moved to the appendix. Please see the revised manuscript. The Section 4 "Results and discussion" is separated into a "Results" section only. We added the discussion to the conclusions and call this section "Discussion and conclusions", now.

From all presented experiments and results I expect to find some indication in the end on how these experiments are ranked with respect to performance, but no such ranking is reached, only a conclusion that experiments indicate in general improved performance compared to the reference experiment.

*Answer:* Ok, the reviewer raises a good point, and we agree that a ranking is necessary, which need to be discussed. We added ranking and performance measures now to the manuscript, which are discussed. For example, the results show that experiment CCLMv4.5 ranks best based on the root-mean-squared-error and bias compared to the GLEAM data set. We also conclude this from the KGE and DAV performance indices. Experiment CCLMv4.5 has an updated algorithm for stomatal resistance and leaf photosynthesis based on the Community Land Model 4.5. This algorithm has a regulation function of night values of stomatal resistance depending on soil water stress function that is more realistic and distinguishes experiment CCLMv4.5 from CCLMv3.5 or CCLMv4.5e.

Thus, there is no balance between all presented details including the statistical analysis and the overall outcome of the results. Therefore, as an overall judgement I must recommend major revision. Both language level and structure, motivation and balance need major improvements in my opinion. The background and motivation for this manuscript as given in the Introduction

is not clear enough as it is written now. For example, in lines 40-42 you state that "the evapotranspiration simulated by ... TERRA-ML ... was found to be systematically underestimated from April to October during the growing season." But you give no reference to this statement, and it is not clear over which region or regions this conclusion refers to. Is it Europe only or also other regions? Are there no publications available where this underestimation is shown? You refer to published evaporation and transpiration fractions, but I miss any comment on how these fractions are estimated by TERRA-ML. Later on (lines 51-52) you state that "plant transpiration is calculated in current version of TERRA-ML with errors (Stockle, 2001)". Here it would be good to also say what kind of errors you mean. I would also say that even if TERRA-ML, now based on empirical stomatal conductance parametrisation, would have given good evapotranspiration in validations of hindcast simulations it can still be motivated to introduce a more advanced stomatal conductance formulation since an empirical formulation may not be valid in changing climate conditions including rising CO2 levels. But if you wish to motivate your work based on bad performance you need to show this bad performance more clearly. Overall, the Introduction section now gives a bit jumpy feeling between very overview style paragraphs (lines 38-38, 53-63, 70-74) and on the other hand very TERRA specific comments (lines 40-42, 48-53, 63-68). Also, all the version details in lines 75-83 do not clarify much. I would recommend that you revise the Introduction to find a better balance between background, TERRA details and motivation for your work.

*Answer:* Ok, we thank the reviewer for this thoughtful comment. Major changes are introduced for the "Introduction" section. We agree that the current TERRA-ML version based on empirical stomatal conductance parameterization gives correct values of evapotranspiration in validations of hindcast simulations. Nevertheless, COSMO generally underestimates evapotranspiration. We added this to the "Introduction" section and added the references (Shrestha and Simmers, 2019, Regenass et al., 2021). But as the reviewer mentioned an introduction of a more advanced stomatal conductance formulation is necessary since an empirical formulation may not be valid in a changing climate including rising $CO_2$ levels. We made this point now more clear in the "Introduction" section. Further, we have rewritten the "Introduction" section and hope that the background, TERRA-ML and motivation of our work have a better balance to the reader now. Please see the revised "Introduction" section in our manuscript.

We rewrote the sentence from lines 40 – 42: "However, the evapotranspiration simulated by the multilayer land surface scheme TERRA-ML of the Consortium for Small scale Modelling – COSMO (http://www.cosmo-model.org/, last access: 09 September 2021) was found to be systematically underestimated from April to October during the growing season" and combined this phrase with the phrase from Line 49. The phrase from Line 49 was deleted. We write now: "Nevertheless, the evapotranspiration simulated by the multilayer land surface scheme TERRA-ML of the Consortium for Small-scale Modelling (COSMO. http://www.cosmo-model.org/, last access: 03 February 2022) was found to be systematically underestimated based on the averaged diurnal cycle of evapotranspiration over Europe during the growing season for the vegetated land surface (Schulz et al., 2015; Shrestha and Simmers, 2019)."

The sentence from the Lines 51 – 52 "However, the plant transpiration is calculated in current version of TERRA-ML with errors (Stockle, 2001), which are related to the simplified parametrization scheme for stomatal conductance ($g_{st}$) or its reciprocal – stomatal resistance ($r_s$). The new one is: One of the possible causes of underestimated evapotranspiration is that

transpiration in TERRA-ML is calculated with inaccuracy due to the simplified stomatal resistance parameterization scheme.

Detailed comments:

***Line 30:*** In my mind, for such a very general statement like "The land surface processes significantly affect the conditions in the low-level atmosphere" it is more appropriate and respectful to refer to well recognized reviews in the area like e.g. Betts et al. 1996 than to one's own very recent paper.

***Answer:*** Ok, thank you for this information, it is a good idea. The added the Betts et al., 1996 work to our manuscript.

***Line 49:*** What do you mean by "not sufficiently represented"? Please be more specific.

***Answer:*** Ok, we agree. The phrase "One of the possible reasons of the underestimation of evapotranspiration is connected with the fact that in TERRA-ML the vegetation is not sufficiently represented in the surface energy balance (Schulz et al., 2015)." was deleted. Also, the sentence from Lines 40 – 42 was deleted. We combined these two deleted phrases and now we have a new one. This sentence is presented in the previous answer to the question about Lines 40 – 42.

***Line 52:*** "Stockle" should be "Savabi and Stockle".

***Answer:*** Ok, Thank you for this comment. We adjusted the reference publication.

***Lines 73-74:*** I find the sentence and statement "However, these schemes have not been implemented into production (exploitation) at convection-permitting scale" a bit strange. Okay, so you mean that dynamic vegetation should be implemented just because it is missing or for some other reason? Please be more specific.

***Answer:*** Ok, the paragraph was rewritten. We think that a more advanced stomatal conductance formulation is necessary since an empirical formulation as it is now implemented in COSMO-CLM may not be valid in a changing climate including rising $CO_2$ levels. The implementation of our new algorithms was guided by the ideas and published materials (e.g.: documentations, model codes) of several existing dynamic vegetation models such as: CARAIB (Dury et al., 2011), Community Land Model (Oleson et al., 2010 and 2013), SURFEX (Le Moigne, 2018) and CHTESSEL (Nogueira et al., 2020). Special attention was paid to the successful examples of the CLM implementation into different regional climate models, for example into the WRF model (Van Den Broeke et al., 2017) or into COSMO-CLM (Davin et al., 2011; Davin and Seneviratne, 2012). The last version is called COSMO-CLM$^2$ and the main focus of this version is coupling to different models, since COSMO-CLM (v4.8) and CLM3.5. In their work (Davin et al., 2011; Davin and Senevirante, 2012) have coupled COSMO-CLM with CLM and found improvements with respect to land surface fluxes, including an improved magnitude of radiation fluxes and a better partitioning of turbulent fluxes, but the multi-layer soil model TERRA-ML used in COSMO-CLM was fully replaced in COSMO-CLM2 with the CLM3.5 parameterization scheme. The COSMO-CLM$^2$ was created and tested, but Davin et al. (2011) did not perform the convection-permitting scale simulations (Prein et al. 2015), due to high computational costs (Stökli et al., 2008 and 2011). All our improvements have been directly implemented in TERRA-ML of COSMO-CLM that allowed us to improve TERRA-ML and save all the advantages of COSMO-CLM (for example, convection-permitting scale). These

changes distinguish our research from the research of (Davin et al., 2011; Davin and Seneviratne, 2012) for coupling the COSMO-CLM and CLM models.

*Line 85:* It is not clear now what "these limitations" exactly refer to. Please be more precise.

*Answer:* Ok, we added more information about limitations (e.g. empirical approach, no connection between carbon dioxide and stomatal resistance, etc.) and change the orders of the sentences in the manuscript making the phrase "these limitations" more appropriate.

*Line 115:* The formulation "atmospheric parameters under the soil" is probably not correct I assume.

*Answer:* Ok, the reviewer raises a good point. The new sentence is: The surface and soil processes are calculated in the multi-layer soil model TERRA-ML (Schrodin and Heise, 2002) consisting of two parts. The first one considers hydrological processes including snow melting and freezing. The second one includes algorithms intended for calculations of bare soil evaporation and plant transpiration, which are computed for non-vegetated and vegetated areas, respectively.

*Line 123:* Hmhm, just wonder if the factor Ld, representing Leaf Area Index, in Eq 58 in Dickinson et al. (1993) is missing here or it is just a different definition of transpiration?

*Answer:* Ok, Thank you for this comment. We added the new, more detailed description of the COSMO-CLM algorithm for transpiration. The new paragraph looks like:

The BATS-based formulation of the plant transpiration is presented in Eq. (1):

$$T_r = f_{plnt} \, (1 - f_i) \, (1 - f_{snow}) \, E_{pot}(T_{sfc}) \, r_a \, (r_a + r_f)^{-1} \tag{1}$$

where $T_r$ is plant transpiration, $r_a$ and $r_f$ are atmospheric and foliage resistance, respectively, $f_{plnt}$, $f_i$, $f_{snow}$ are fractional areas covered by plants, intercepted water, and snow, $E_{pot}(T_{sfc})$ is potential evapotranspiration. In accordance with the official documentation of COSMO-CLM model (Doms et al., 2018) stomatal resistance is a part of foliage resistance, which is equal to:

$$r_f^{-1} = r' C_F \tag{2}$$

where $r'$ is reduction of transpiration by stomatal resistance equal to $r' = r_{la}(r_{la} + r_s)^{-1}$, $C_F$ is parameterized by $C_F = f_{LAI} \, r_{la}^{-1}$. $f_{LAI}$ – is the leaf area index, The detailed formulations of stomatal resistance algorithms are presented in next subsection 2.2.

**Line 124:** In Eq 1 it says "Tr" but here "Trk". Please make it consistent.

*Answer:* Ok, we agree. The parameter name was corrected to "Tr".

*Lines 169-171:* Although it is very precise to divide the text in "Current" and "New" formulation subsections it is from a stylish perspective a bit awkward when the "Current" section is represented by only one sentence. Therefore, I would recommend to remove the subsections here. The same comment is valid for Section 2.4.

*Lines 184-187:* See comment for lines 169-171.

*Answer:* Ok, Thank you for this note. We think that this division the Method section on "Current" and "New" formulations is important and allows to understand more clearly the

differences between methods. We also agree with the Reviewer, and we updated the section "Current". *The new subsubsection "Current" for leaf photosynthesis is:*

In the current model version of TERRA-ML, there are no algorithms for estimating leaf photosynthesis. In the reference version of COSMO-CLM model, this algorithm is not needed for calculations and plants are represented by the following vegetation parameters, which are read in by the model as external 2D fields coming from remote sensing data. The vegetation parameters, which are read in, are leaf area indexes, plant coverage, minimum stomatal resistance, root depth and roughness length.

*The new subsubsection "Current" for radiation fluxes is:*

In the current version of COSMO-CLM a canopy layer is presented as a "one-big leaf". In this approach, all leaves of the canopy have the same plant physiological properties and relative responses to the environment as any single unshaded leaf in the upper canopy. Additionally, in COSMO-CLM there are several assumptions simplifying this approach. The first one is water vapor flux between the plant foliage and the canopy air is equal to the flux between air inside and air above the canopy ($Tv = Tg$). The second one is the foliage temperature to be equal to the surface temperature (Doms et al., 2018).

***Line 196, Eq 8:*** Please replace "sun" with "sha".

***Answer:*** Ok, we agree. The equations were corrected

***Lines 287-292:*** Very complicated paragraph where I assume the main message is simply "Gridded observational data sets (E-OBS, HYRAS, GLEAM) were interpolated to the COSMO-CLM grid for comparison.", right?

***Answer:*** Ok, we thank the reviewer for this thoughtful comment. The paragraph was corrected the new one is: As an additional instrument for validating model results with the new formulations, we compared COSMO-CLM results with the gridded observational data sets. It allowed us to get more precise statistical scores because of the models and gridded observational data sets represent average values than processes in specific points (Osborn and Hulme, 1998). In the analysis, we used the gridded data sets with information about precipitation, temperature, and evaporation for validation of COSMO-CLM parameters. The gridded observational data sets (E-OBS, HYRAS, GLEAM) were interpolated to the COSMO-CLM grid for comparison.

***Lines 316-349:*** I don't see the point to spend a considerable part of the discussion on how values look for the inactive vegetation periods (wintertime and night-time). In my mind the most interesting part is how they differ during summer daytime. But this part cannot be analysed by these figures since one cannot distinguish any differences due to the y-axis scale. I would recommend to focus your analysis more on the summer daytime part.

***Answer:*** Ok, we thank the reviewer for this thoughtful comment, we agree with it. All stomatal resistance plots were recalculated in accordance with this comment. The figures 2, 3 were updated. In addition, we have added additional information about stomatal resistance data from TRY database over Germany. The other plots are also updated with the main focus on changes during summer months.

***Lines 351-371:*** You start the paragraph by concluding that "stomatal resistance ... is a highly intermittent phenomenon, extremely localized on the leaf level, and varies with leaf positioning on a plant and from leaf to leaf and from plant to plant" but then you compare your model results with observations from literature based on "located in the North America region" with no further comments on if these observations can at all be considered to be representative for your model results. Thus, this first sentence and your final comparison does not make sense to me.

*Answer:* Ok, the reviewer raises a good point in these 2 comments. We agree that the main focus in Section 4.1 should be shifted to the analysis of summer day-time values of stomatal resistance, because of that all plots were updated, table 3 – recalculated. The text of the section was rewritten. Also, we added more information about published data from North America and we assume that our experimental stomatal resistance data for $C_3$ grass can be compared with the in-situ data published earlier due to several causes: 1) Vegetation in published data is presented by grass and includes the *Lolium perenne* species; 2) The North American regions presented in the research situated similar climate conditions. The phrase "stomatal resistance ... is a highly intermittent phenomenon, extremely localized on the leaf level, and varies with leaf positioning on a plant and from leaf to leaf and from plant to plant" was corrected and the new phrase is "Stomatal resistance validation of the reference and experimental results presented in timeseries format is a formidable task. Due to measuring stomatal resistance (conductance) is a resource-intensive task, especially for its continuous quantification over time and there are no long in-situ time series or datasets including daily stomatal resistance data."

***Lines 374-421*** (Section 4.2 and Figures 4-5): The comparison between model results and GLEAM datasets in Figure 4 shows that the difference between the GLEAM datasets are often as big or bigger than the differences between the model versions, especially for AEVAP. Thus, in my mind it is difficult to draw any further conclusions from this comparison other than perhaps that ZVERBO for the new model versions is better than CCLMref. The statistical analysis with all numbers presented is not necessary to reach this conclusion I would say. And the analysis gives no indication on which of the new model versions are better or worse, right?

*Answer:* Ok, thank you for this comment. The all experiments were recalculated, the figures replotted. The main focus changed to the analysis of changes between the algorithms from May to September.

***Lines 423-437*** (Section 4.3 and Figure 6): As for the section on "Evapotranspiration and evaporation" the statistical analysis with all detailed numbers of sensible and latent heat fluxes is not needed to reach your conclusion (visible from the figure) that "experiment results are similar to the CCLMref data". Thus, in my mind unnecessary long details for this conclusion.

*Answer:* Ok, thank you for this comment. Yes, it is true that difference in latent and sensible heat fluxes between the experiments and the reference is small. Nevertheless, the experiments show slightly better results, and we think that it should be demonstrated in the manuscript. However, this Section 4.3 was relocated to the Appendix. We corrected the text in the Results section in accordance with the section results. Also, the text and figures of this section were rewritten and replotted. We spend more attention on the analysis during the months May to August, where vegetation is in its active phase.